



# Future streamflow regime changes in the United States: assessment using functional classification

Manuela I. Brunner[1], Lieke A. Melsen[2], Andrew J. Newman[1], Andrew W. Wood[3,1], and Martyn P. Clark[4]

[1]National Center for Atmospheric Research (NCAR), Boulder CO, United States
[2]Hydrology and Quantitative Water Management, Wageningen University, Wageningen, Netherlands
[3]Climate and Global Dynamics Laboratory, National Center for Atmospheric Research (NCAR), Boulder CO, United States
[4]College of Arts and Science, University of Saskatchewan, Canmore, Canada

**Correspondence:** Manuela I. Brunner (manuelab@ucar.edu)

**Abstract.** Streamflow regimes are changing and expected to further change under the influence of climate change with potential impacts on flow variability and the seasonality of extremes. However, not all types of regimes are going to change in the same way. Climate change impact assessments can therefore benefit from identifying classes of catchments with similar streamflow regimes. Traditional catchment classification approaches have focused on specific meteorological and/or streamflow indices usually neglecting the temporal information stored in the data. The aim of this study is two-fold: (1) develop a catchment classification scheme that allows for the incorporation of such temporal information and (2) use the scheme to evaluate changes in future flow regimes.

We use the developed classification scheme, which relies on a functional data representation, to cluster a large set of catchments in the conterminous United States (CONUS) according to their mean annual hydrographs. We identify five regime classes that summarize the behavior of catchments in the CONUS: 1) *Intermittent regime*, 2) *weak winter regime*, 3) *strong winter regime*, 4) *New Year's regime*, and 5) *melt regime*. Our results show that these spatially contiguous classes are not only similar in terms of their regimes, but also their flood and drought behavior, as well as their physiographical and meteorological characteristics. We therefore deem the functional regime classes valuable for a number of applications going beyond change assessments including model validation studies or the prediction of streamflow characteristics in ungauged basins.

To assess future regime changes, we use simulated discharge time series obtained from the Variable Infiltration Capacity hydrologic model driven with meteorological time series generated by five general circulation models. A comparison of the future regime classes derived from these simulations with current classes shows that robust regime changes are expected only for currently melt-influenced regions in the Rocky Mountains. These changes in mountainous, upstream regions may require the adaptation of water management strategies to ensure sufficient water supply in dependent downstream regions.

**keypoints:**

1. Functional data clustering enables forming clusters of catchments with similar hydrological regimes and a similar drought and flood behavior.





2. We identify five streamflow regime clusters: 1) Intermittent regime, 2) weak winter regime, 3) strong winter regime, 4) New Year's regime, and 5) melt regime.

3. Future regime changes are most pronounced for currently melt-dominated regimes in the Rocky Mountains.

4. Functional regime clusters have widespread utility for predictions in ungauged basins and hydroclimate analyses.

**keywords:** functional data analysis, clustering, climate change, prediction in ungauged basins, CAMELS, random forest

# 1 Introduction

The characteristics of streamflow regimes, as described by mean annual hydrographs, govern the hydrological functioning of a catchment by influencing streamflow variability and seasonality and the timing of extremes such as droughts and floods. Such regimes are undergoing changes and expected to further change under future climate conditions (Addor et al., 2014; Arnell, 1999; Brunner et al., 2019b; Horton et al., 2006; Laghari et al., 2012; Leng et al., 2016; Milano et al., 2015). Regime changes are caused by changes in precipitation seasonality and intensity (Brönnimann et al., 2018) and seasonal shifts and decreases in melt contributions (Farinotti et al., 2016; Jenicek et al., 2018) related to reduced snow and glacier storage (Beniston et al., 2018). Predicted regime changes are relatively robust (Addor et al., 2014) compared to changes in high and low flows, which are highly uncertain (Brunner et al., 2019c; Madsen et al., 2014) because of diverse uncertainty sources introduced in various steps along the modeling chain (Clark et al., 2016). It has been shown that future regime changes can be linked to changes in flood and drought characteristics, e.g. the seasonality and magnitude of floods (Middelkoop et al., 2001) or the duration of droughts (Brunner and Tallaksen, 2019). Quantifying hydrological regime changes can assist in inferring changes in extremes and is crucial for adapting water management practices (Clarvis et al., 2014).

We can improve our understanding of regime changes by employing regime classification in climate change impact assessments (Coopersmith et al., 2014). Most existing (regime) clustering approaches focus on a set of indices either referring to certain physiographical or climatological catchment characteristics (Knoben et al., 2018; Wolock et al., 2004), specific streamflow indices (Archfield et al., 2014; Bower et al., 2004; Haines et al., 1988; McCabe and Wolock, 2014), or a mixture of the two (Berghuijs et al., 2014; Coopersmith et al., 2012; Kuentz et al., 2017; McManamay and Derolph, 2019; Sawicz et al., 2011; Sharghi et al., 2018; Wagener et al., 2007). The use of catchment characteristics can be problematic because there is often no clear link between these characteristics and streamflow indices (Ali et al., 2012; Addor et al., 2018). One may therefore prefer to work with streamflow indices directly when identifying catchment classes with a similar streamflow behavior. However, the use of streamflow indices requires the subjective choice of streamflow indices which may not fully capture the catchment behavior. Both the catchment and climate characteristics and the streamflow index approaches neglect nearly all available temporal information embedded in a streamflow time series or regime in the form of temporal (auto-) correlation. Only very few studies have tried to take account of temporal information in clustering hydrological catchments, e.g. by using the shape of the autocorrelation function as an index (Toth, 2013), even though such information is potentially very useful. We here explore





how we can make better use of the seasonal and temporal information stored in the hydrological regime using a functional data
    representation going beyond the consideration of a set of indices.

    In contrast to classical multivariate data, functional data are continuously defined (Ramsay and Silverman, 2002). Functional
    data analysis represents each hydrological regime as a function and therefore circumvents the choice of individual hydrograph
    characteristics, which allows for the exploitation of the full information stored in the time series or annual hydrograph when
clustering catchments (Chebana et al., 2012; Ternynck et al., 2016). The functional form of the data is derived from discrete
    observations (Ramsay and Silverman, 2002) either by smoothing the data non-parametrically (Jacques and Preda, 2014) or by
    projecting the data onto a set of basis functions. The basis function (e.g. B-spline, Fourier, or wavelet bases) coefficients can be
    used for clustering (Cuevas, 2014). It has been shown in previous studies that functional data representations can be beneficial
    in the identification of groups of similar hydrographs over a range of temporal scales, such as spring flood events (duration of
six months; Ternynck et al., 2016), flood events (duration of several days; Brunner et al., 2018), low flow events (Laaha et al.,
    2017), diurnal discharges (duration of one day; Hannah et al., 2000), and yearly hydrographs (Merleau et al., 2007; Jamaludin,
    2016).

    These previous studies focused on a limited number of stations and on current climate conditions. The goals of this study are
    therefore two-fold: (1) to develop a catchment classification scheme for streamflow regimes useful in climate change impact
assessments; and (2) to use this scheme to evaluate changes in future flow regimes.

    We develop the catchment classification scheme for a large dataset of 671 catchments over the United States (Newman
    et al., 2015; Addor et al., 2017) using a functional representation of mean annual hydrographs. This scheme makes better use
    of the seasonal and temporal information stored in the hydrological regime than index-based approaches. In order to assess
    future regime changes, we use streamflow time series simulated with the hydrological Variable Infiltration Capacity (VIC)
model driven by meteorological data derived from five general circulation models (GCMs) under a high emission scenario.
    We compare current and future regime class memberships to identify catchments with future regime changes. Such change
    assessments are of paramount importance in preparing for future water management strategies because future regime shifts can
    influence the variability and timing of high and low flows.

## 2   Data and Methods

### 2.1   Data

We form regime clusters, i.e. clusters of catchments with similar mean annual hydrographs, using observed streamflow data of
671 catchments in the conterminous United States (CONUS) (Newman et al., 2015) with minimal human impacts. The data
were downloaded for the period 1981–2018 from the USGS website https://waterdata.usgs.gov/nwis (R-package dataRetrieval;
De Cicco et al., 2018). In contrast, the regime change analysis uses streamflow simulated by the hydrological Variable Infil-
tration Capacity (VIC) model for a subset of 605 catchments, for which satisfactory model performance could be achieved
    through systematic sampling of parameter sets (Melsen et al., 2018). Physiographical and meteorological characteristics for




these catchments are available via the Catchment Attributes and MEteorology for Large-sample Studies data set (CAMELS) (Addor et al., 2017).

## 2.2 Regime clustering and classification

Hydrological regime clusters are derived using functional data analysis on the observed hydrological regimes of the 671 catchments (Fig. 1). In the functional data framework, each hydrological regime is considered to be a function (Ramsay and Silverman, 2002). To achieve such a functional data representation, we project the discrete observations, i.e. the mean annual hydrographs at daily resolution, to a set of B-spline basis functions (R-package fda; Ramsay et al., 2014) (see illustration in Figure 1.1 a–c) because B-splines are able to mimic the main characteristics of hydrological regimes (Brunner et al., 2018).
A (smoothing) spline function is defined by its order of polynomial segments and its number and placement of knots. The number of knots determines the ability of spline functions to represent sharp features in a curve and the knots can be placed such that they are denser in areas with stronger variations than in smooth areas (Höllig and Hörner, 2013). We here use five spline basis functions of order four, which corresponds to a minimal number of basis functions which still allows for flexibility in representing diverse shapes of regimes. The projection of the observed regimes to the five basis functions results in five
coefficients per observed regime, one per spline base.

The clustering into regime classes is performed using a Euclidean distance matrix computed using the matrix of $n = 671 \times 5$ spline coefficients (Figure 1.2 a–b). We use a hierarchical clustering algorithm allowing for non-elliptical clusters (Gordon, 1999) with Ward's minimum variance criterion, which minimizes the total within-cluster variance (Ward, 1963). To identify an optimal number of clusters, we cut the tree at $k = 2, ..., 30$ clusters and compute the mean silhouette width (Rousseeuw,
1987), which provides a measure of clustering validity, for the different numbers of clusters. We finally determine five regime clusters because the mean silhouette width values stabilize at five clusters. A comparison with regime clusters derived by $k$-means clustering shows that the final clusters formed are relatively stable independent of the choice of the clustering technique. Each of the clusters can be summarized by its median regime identified using the $h$-mode depth which allows for ordering the regimes within a cluster (Cuevas et al., 2007).

To assess whether the similarities of the catchments within a cluster go beyond their regime type, we compare their physiographical (latitude, area, elevation), climatological (mean precipitation, fraction of snow, aridity), and flood and streamflow drought characteristics. The flood and drought characteristics are determined using a peak-over-threshold (Lang et al., 1999) and a threshold-level approach (Yevjevich, 1967), respectively. The flood threshold is fixed at the 25th percentile of the annual maxima time series of each catchment separately to guarantee a balanced number of extracted events across catchments (Schlef
et al., 2019). The drought threshold is fixed at the highest value of the annual minimum time series and the time series smoothed over a window of 30 years to limit the extraction of dependent events (Brunner et al., 2019d; Tallaksen and Hisdal, 1997).





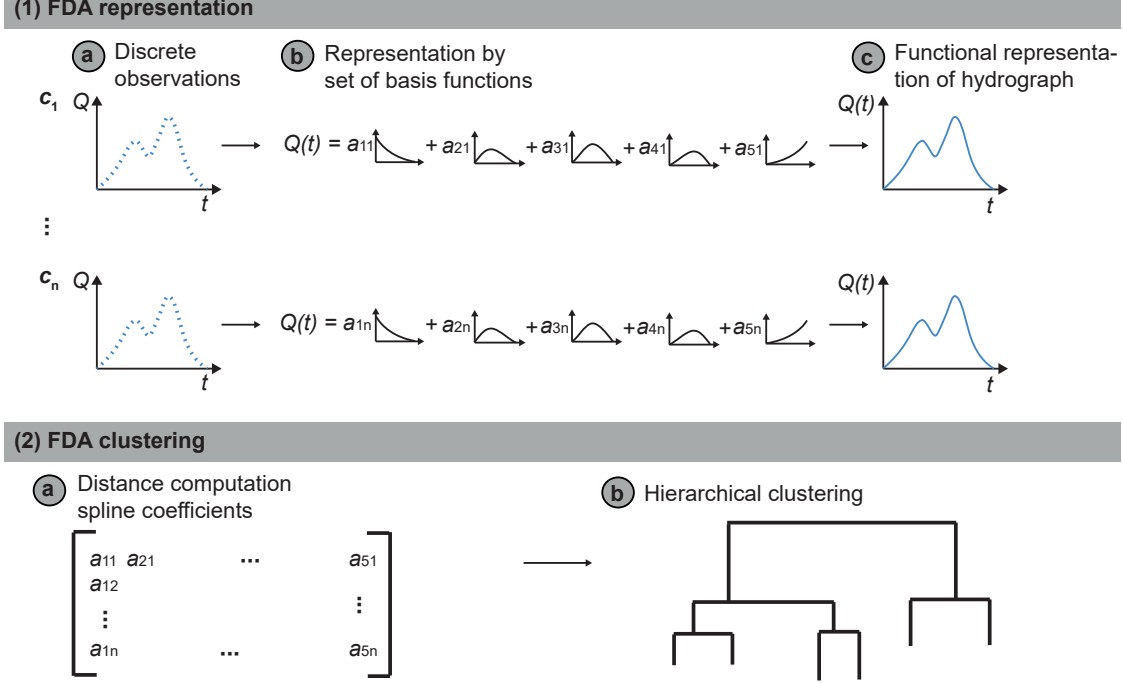

**Figure 1.** Functional data (FDA) clustering procedure: (1) FDA representation of regimes by projecting (a) discrete observations to (b) a set of spline bases to derive a (c) functional representation of the hydrological regimes. (2) FDA clustering by computing (a) a distance matrix using the spline coefficients from Step 1 in a (b) hierarchical clustering procedure.

To further investigate the physiographical and climatological controls on regime class membership, we perform a random forest classification (Breiman, 2001; Harrell, 2015; James et al., 2013). We fit the model to 33 non-hydrological catchment characteristics in the CAMELS dataset (Addor et al., 2017), i.e. topographical, soil, geological, and climatological characteristics, excluding gauge ids and characteristics with missing values, to predict regime class membership (R-package randomForest; Liaw and Wiener, 2002). The related analysis of estimated variable importance allows for identifying factors important in determining regime class membership, which is useful for ungauged basins where the regime class cannot be determined based on discharge observations.

### 2.3 Model simulations

For the regime change analysis, we use daily streamflow time series simulated by Melsen et al. (2018) in a model intercomparison project. They ran the Hydrologiska Byråns Vattenbalansavdelning model (HBV; Bergström, 1976), the Variable Infiltration Capacity model (VIC; Liang et al., 1994), and the Sacramento Soil Moisture Accounting model (SAC-SMA) combined with SNOW–17 (Newman et al., 2015) for 605 catchments in the CAMELS dataset for a period representing current (1985–2008) and future climate conditions (2070–2100). Each of these models was run with a large number of parameter sets sampled using Sobol-based Latin hypercube sampling (Bratley and Fox, 1988) and forced with daily observed meteorological variables for





the current period (Daymet; Thornton et al., 2012). The performance of each of the sampled parameter sets was evaluated by comparing the model simulations with observed discharge data over a 23-year period (1985–2008) (USGS, 2019) using the Kling–Gupta efficiency metric (Gupta et al., 2009) defined as:

$$E_{KG}(Q) = 1 - (\rho - 1)^2 + (\alpha - 1)^2 + (\beta - 1)^2, \tag{1}$$

where $\rho$ is the correlation between observed and simulated runoff, $\alpha$ is the standard deviation of the simulated runoff divided by the standard deviation of observed runoff, and $\beta$ is the mean of the simulated runoff, divided by the mean of the observed runoff.

Here we focus on the VIC model and those model runs derived using the parameter set resulting in the best model performance in terms of $E_{KG}$. $E_{KG}$ values over all stations ranged from a first quartile of 0.47 over a median of 0.60 to a third 140 quartile of 0.71 with the lowest values obtained in the Great Plains.

Melsen et al. (2018) forced the VIC model with daily output from General Circulation Models (GCMs) which was statistically downscaled using the bias-correction and spatial disaggregation (BCSD) method of Wood et al. (2004), for both the current and future period (Department of the Interior, Bureau of Reclamation, Technical Services Center, 2013). They used the output of five different climate models from the Coupled Model Intercomparison Project Phase 5 (CMIP5; Taylor et al., 2012) 145 including CCSM4 (ccsm), CNRM-CM5 (cnrm), INM-CM4 (inmcm), IPSL-CM5A-MR (ipsl), and MPI-ESM-MR (mpi), and the Representative Concentration Pathway 8.5 (RCP8.5; Moss et al., 2010) representing a high emission scenario. We use three types of model runs; a control run, where the hydrological model is forced with the observed Daymet meteorology (1985–2008); five reference runs, one per GCM, where the hydrological model is forced with the simulated meteorology for current conditions (1985–2008); and five future runs, where the hydrological model is forced with simulated meteorology for the future 150 period (2070–2100). We refer to the regimes derived from the control run as the control regimes, those regimes derived from the reference simulations as the reference regimes, and those regimes derived from the future runs as future regimes.

## 2.4 Evaluation of simulated regimes

To determine the suitability of the VIC model for representing regime changes, we extend the model evaluation from the Kling-Gupta efficiency $E_{KG}$ (Eq. 1), which provides an integrative measure of model performance, to a climate sensitivity 155 analysis performed on the control run and a comparison of observed and simulated regime classes performed on the control and reference runs. In the climate sensitivity analysis, we assess whether the hydrological model reacts to changes in mean temperature and precipitation in the same way as observations. To do so, we follow a technique presented in Wood et al. (2004) that involves creating many samples of modeled and observed climate and streamflow, and assessing sensitivities from mean behavior of each sample. The multi-year samples help to average out the confounding effects of other influences, such 160 as the initial catchment moisture in individual years. Accordingly, we generate new temperature, precipitation, and streamflow time series by resampling the available hydrological years with replacement ($n = 5000$ times). We compute mean temperature, precipitation, and streamflow for the resampled time series to derive a relationship between mean streamflow and the two





meteorological variables. Conducting this experiment for both observed and simulated time series supports analysis of whether the simulated streamflow time series react to changes in mean annual climate in the same way as observed time series.

To assess the ability of the VIC model to simulate the observed regime class, we compare observed to simulated regime classes for the control and reference runs. To assign simulated regimes to one of the five classes, we fit a second classification model using a random forest, which allows for the classification of a given mean annual hydrograph into one of the five regime classes using its B-spline coefficients. This analysis is different from the first random forest analysis, which was aimed at identifying catchment and climate characteristics determining class membership. We use 10-fold cross validation (Hastie et al.,

2008) to evaluate the capability of the classification model to correctly predict observed regime classes. The cross-validation shows that the regime-class prediction error is only 2% and that the model can be used for the accurate prediction of class memberships of simulated regimes. We compare the observed regime classes to the regime classes predicted with the random forest model for the simulated control regimes. This comparison shows that the VIC model is well capable of simulating hydrological regimes with a correct regime prediction in more than 95% of the catchments. The prediction error roughly

doubles when using the reference instead of the control regimes indicating that additional uncertainty is introduced by using the GCM simulations as meteorological forcing.

## 2.5   Future regimes

We use the hydrological model simulations to assess regional changes in regime class memberships. To do so, we predict the regime classes for the five reference regimes (one per GCM) and the corresponding future regimes using the random forest

classification model. We then compare the predicted future class to the class of the corresponding reference simulation. We look at the (dis-) agreement of predicted regime changes for the five GCMs and evaluate whether and where most models agree on regime change.

## 3   Results and Discussion

### 3.1   Hydrological regime clusters

Based on the functional data clustering, the hydrological regimes of the 671 catchments in the CAMELS dataset are divided into five clusters resulting in five mostly spatially contiguous regions of catchments with similar annual hydrographs (Fig. 2).





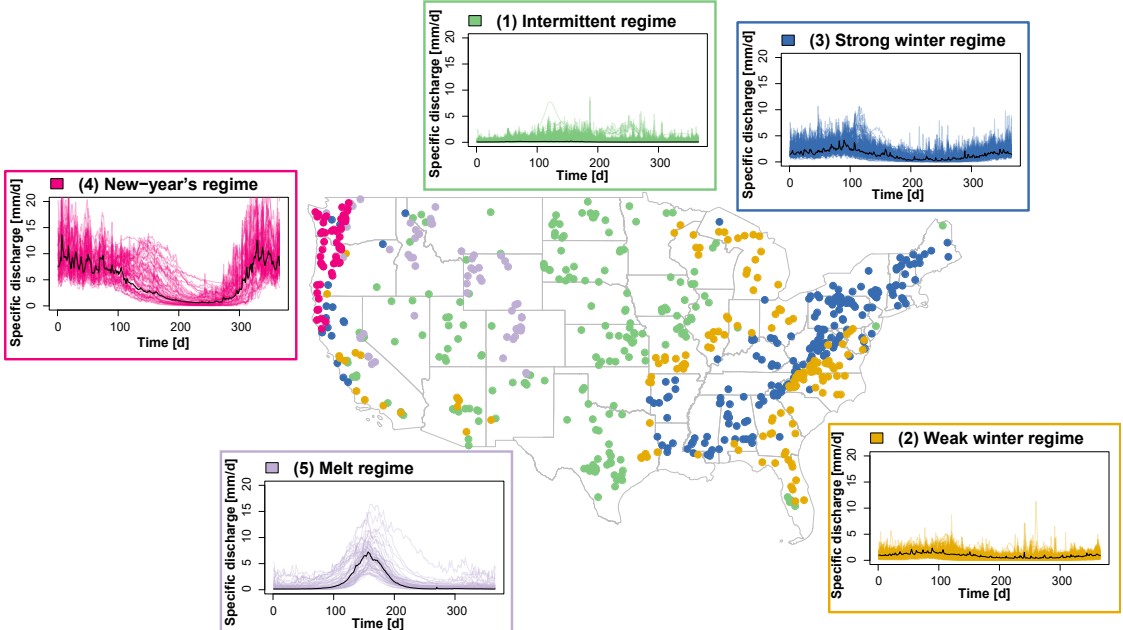

**Figure 2.** Map of regime clusters and the regimes of the catchments belonging to the five clusters: 1) Intermittent regime, 2) weak winter regime, 3) strong winter regime, 4) New Year's regime, and 5) melt regime. Regimes of individual catchments are colored according to their cluster membership and the median hydrograph per cluster is given in black.

1. The first cluster, which we here call the *intermittent regime* cluster, comprises regimes with a very weak seasonality, dominated by the occurrence of short precipitation events related to thunderstorms or fronts. The catchments belonging to this region mostly lie in the Great Plains, the Great Basin, and the Plateau region (158 catchments).


2. The second cluster, here referred to as *weak winter regime*, comprises regimes showing a weak seasonality with slightly more discharge in winter and spring than in summer and fall. The catchments belonging to this cluster lie in the Coastal Plain, the Lake region, and parts of the Prairie region (127 catchments).

3. The third cluster, i.e. the *strong winter regime*, is similar to the previous regime type with higher winter and spring discharge compared to summer and fall but a slightly more expressed seasonality. The catchments in this cluster mostly

belong to the Appalachian region (206 catchments).

4. The catchments in the fourth cluster, which we call *New Year's regime*, have a very strong seasonality with high discharge in winter in general and around New Year in particular, but low discharge in summer. Catchments in this region are located in the Pacific Northwest (57 catchments).

5. The fifth cluster comprises regimes that are snowmelt-dominated and show high discharge in spring and summer vs.

low discharge in winter and fall. The catchments belonging to this *melt regime* are located in the Rocky Mountains (57 catchments).





The regime classes are provided for the 671 catchments in the CAMELS dataset via HydroShare (Brunner, 2020).

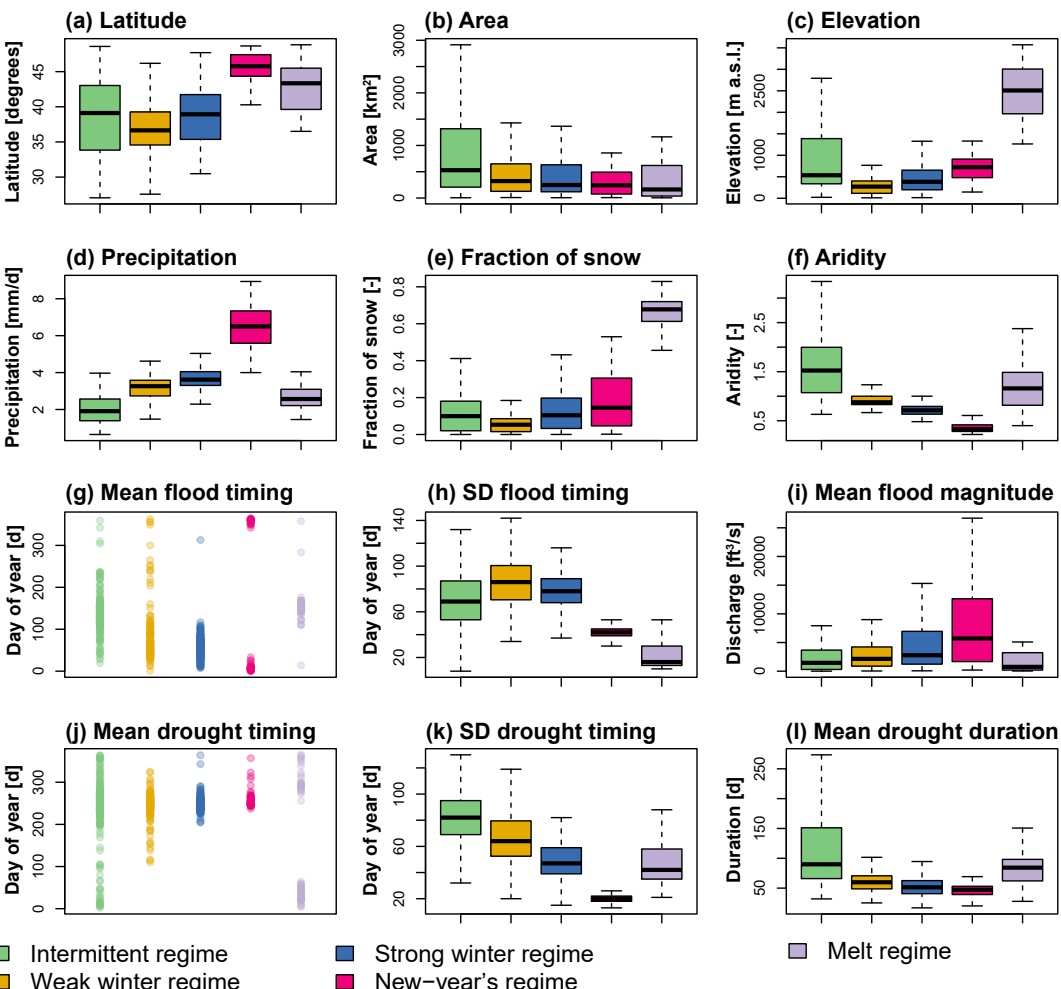

**Figure 3.** Catchment (a–f), flood (g–i), and drought characteristics (j–l) of the catchments belonging to the five regimes: intermittent, weak winter regime, strong winter regime, New-year's, and melt. Characteristics (a–f) were derived from the CAMELS dataset, the flood and drought characteristics using a peak-over-threshold/threshold-level approach, respectively. SD = standard deviation.

The catchments in the five regime clusters are not only similar in terms of their regimes, according to which the clusters were formed, but also in terms of their physiographical, climatological, and flood and drought characteristics (Fig. 3).

Catchments with an *intermittent regime* are comparably large, receive only small precipitation amounts, and are dry. Floods occur mainly in spring and summer while droughts occur in fall and winter. Flood magnitudes are comparably small while droughts are longer than droughts of catchments belonging to other regime clusters. Catchments with a *weak winter regime* lie at low elevations and only a small fraction of total discharge is contributed by snow. These catchments show flood occurrence in winter and spring and droughts in fall. Catchments with a *strong winter regime* lie at relatively low elevations and receive a





medium amount of precipitation. Floods occur in winter and droughts in fall. Catchments with a *New Year's regime* lie at high latitudes and receive a lot of precipitation. Floods occur around New Year and droughts in late fall. Flood magnitudes are very pronounced. Catchments with a *melt-dominated regime* lie at high elevations and a large part of their discharge is melt water. Floods in these catchments occur in spring and early summer due to melt processes and droughts occur in the winter months due to snow accumulation.

The random forest classification model fitted to the regime clusters and a variety of physiographical and climatological catchment characteristics allows for a reliable prediction of the correct regime class (prediction error 10%) based on catchment characteristics only. The related variable importance analysis shows that the the most important variables for predicting regime classes are climatological characteristics including mean precipitation and aridity. Important physiographical predictors include the longitude and latitude of the gauge location and catchment mean slope and elevation.

We find functional data clustering to be a useful tool for identifying clusters of catchments with not only similar streamflow regimes but also similar catchment, meteorological, flood and drought characteristics. The five regime clusters are mostly spatially contiguous and show similarities to the four catchment clusters built by McManamay and Derolph (2019) who used 110 different hydrological characteristics in their clustering procedure. Our approach circumvents the computation and selection of (a large number of) streamflow characteristics by applying the clustering procedure on a functional representation of the mean 225 annual hydrographs directly. The five regime clusters identified also show spatial similarities with the ten catchment clusters formed by Jehn et al. (2019) for the same set of catchments using a small set of hydrological streamflow characteristics. However, our clustering scheme avoids the formation of very small clusters seen in Jehn et al. (2019). Similarly to Jehn et al. (2019) and Yaeger et al. (2012), we find that meteorological characteristics in general and mean precipitation and aridity in particular are better predictors for hydrological class membership than physiographical catchment characteristics. However, we also find 230 that catchment mean slope, elevation, and location help to explain regime class membership. The strong link between regime classes and meteorological and physiographical catchment characteristics allows for the attribution of ungauged catchments, where streamflow data are not available, to one of the regime classes, which is potentially very useful for the prediction of streamflow characteristics in ungauged basins.

## 3.2 Model validation

Before simulations are used to investigate changes in streamflow regimes, we tested whether climate sensitivity is realistically mimicked by the applied model. The simulated time series show a similar reaction of mean discharge to changes in mean temperature and precipitation as the observed series (Fig. 4).





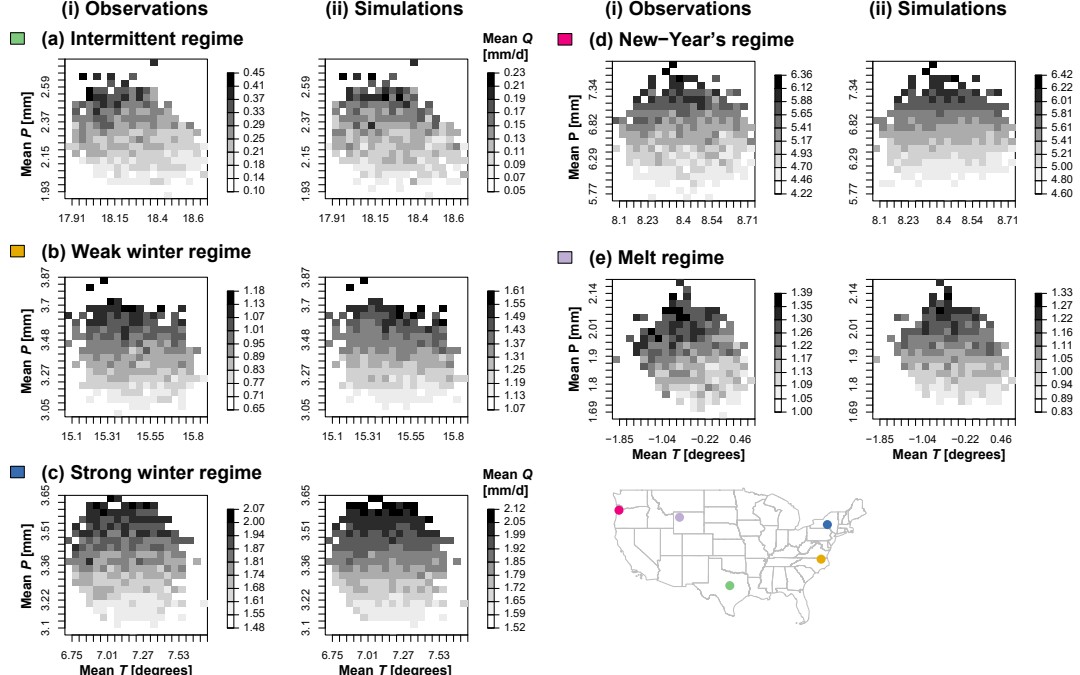

**Figure 4.** Climate sensitivity analysis for observations (i) and simulations (ii): Dependence of mean discharge ($Q$) on precipitation ($P$) and temperature ($T$) for five example catchments, one per regime class: (a) intermittent regime: Cowhouse creek at Pidcoke, TX (USGS 08101000); (b) weak winter regime: Potecasi creek near Union, NC (USGS, 02053200); (c) strong winter regime: Otselic river at Cincinnatus NY (USGS 01510000); (d) New Year's regime: Tucca creek near Blaine, OR (USGS 14303200); and (e) melt regime: South Fork Shoshone river near Valley, WY (USGS 06280300).

Higher mean precipitation leads to higher mean discharge independent of the catchment and regime. The reaction of stream-flow to temperature, however, seems to depend on the catchment because the relationship between mean temperature and mean

discharge is generally weak and can be positive or negative. Based on a visual analysis, the realistic simulation of climate sensitivities of mean discharge by the VIC model make it a suitable choice for climate impact assessments of regimes. A quantitative comparison of gradients in these response surfaces over all catchments confirms that the observed and modeled temperature sensitivities are weak while precipitation sensitivities are similar (Klomogorov–Smirnov test not rejected at level of significance $\alpha = 0.05$).

The VIC model is also able to simulate regimes matching the observed regime classes. The classes of the simulated control regimes predicted using the random forest classification model match the observed regime classes in more than 95% of the catchments (prediction error <5%). The regime class prediction error almost doubles for the reference regimes (prediction error 8–10%) but still allows for the simulation of the correct regime class in more than 90% of the catchments. The good match of simulated control and reference regimes with the observed regimes is illustrated in Fig. 5 for the example catchments with a

weak and strong winter regime, a New-Year's, and a melt regime (b–e). In contrast, the regime of the example catchment with an intermittent regime is poorly simulated (a).

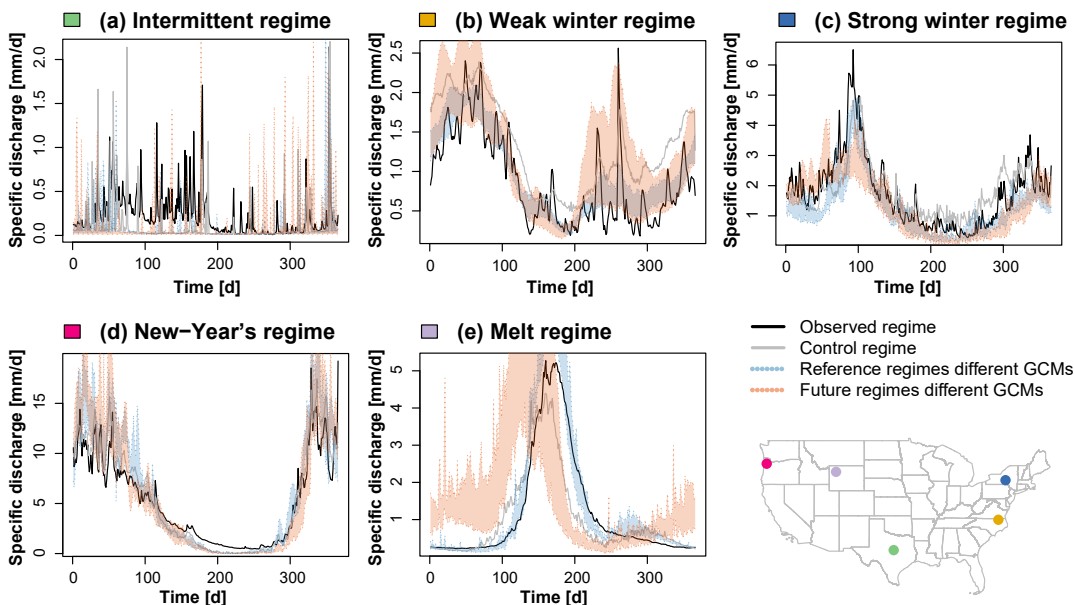

**Figure 5.** Comparison of observed (black) and simulated control regimes (observed meteorology; grey) with simulated reference (1981-2008; blue) and future regimes (2070-2100; red) derived from the five GCMs for the five example catchments, one per regime type: (a) intermittent regime: Cowhouse creek at Pidcoke, TX (USGS 08101000); (b) weak winter regime: Potecasi creek near Union, NC (USGS, 02053200); (c) strong winter regime: Otselic river at Cincinnatus NY (USGS 01510000); (d) New Year's regime: Tucca creek near Blaine, OR (USGS 14303200); and (e) melt regime: South Fork Shoshone river near Valley, WY (USGS 06280300).

The results of our model evaluation show that the VIC model performs well in simulating the correct regime types when forced with observed meteorological data and in simulating changes in mean discharge as a response to changes in mean temperature and precipitation. However, simulating the observed regime classes becomes more difficult when forcing the

model with simulated meteorological data generated by GCMs, in particular in certain areas in the Midwest, in the Pacific Northwest, and a few catchments in the Rocky Mountains and Florida. Over all catchments, regimes of catchments with a weak winter regime and an intermittent regime, i.e. regimes with a weak seasonality are not well reproduced in GCM-forced simulations (Fig. 6 left bars). In contrast, regimes with a strong seasonality such as strong winter and New Year's regimes are well simulated. These results highlight that model performance depends on regime type. The uncertainty introduced by

using the GCM meteorology as shown by differences between the downscaled and observed time series could have different reasons. One potential reason for these differences is that the observations used to fit the downscaling model are fairly short. Another reason could be that the downscaling model was fitted using a different dataset (Maurer et al., 2002) than used for





the calibration of the hydrological model (Thornton et al., 2012) highlighting that precipitation observations are subject to measurement errors.

## 3.3 Future regime simulations

Our results show that streamflow regimes may be subject to future changes. This is illustrated by the regime shift of the catchment with a melt regime in Figure 5e. However, these regime shifts do not affect all catchments and are to some extent dependent on the GCM and regime considered (Fig. 6). Only few regime changes are expected for catchments with a currently intermittent, strong winter, and New Year's regime. Moderate regime changes are predicted for catchments with a currently

weak winter regime, however, simulation error is quite large for this type of regime. The biggest changes are predicted for currently melt-dominated regimes while catchments with current New Year's regimes hardly change. Currently intermittent regimes are mostly changing to weak winter regimes, currently weak winter regimes to intermittent or strong winter regimes, and currently strong winter regimes to weak winter or New Year's regimes, regime types relatively close to their current regime. In contract, melt regimes can change into any type of regime depending on the local climate. Catchments without predicted

regime changes may still undergo changes in individual streamflow characteristics such as variability or low and high flows.

Geographically, regime changes are expected according to most GCMs in the Rocky and Appalachian Mountains and to a lesser degree in the Pacific Northwest and the Midwest. In contrast, regimes of catchments in the Great Plains are predicted to be mostly unaffected by changes. These results are summarized in Figure 7a where all catchments with at least one GCM predicting future regime changes are colored according to their current regime type. Even if all GCMs agree on changes, they

may not agree on the direction of change (Fig. 7b). Catchments where models agree both on changes and their direction are mostly located in the Rocky Mountains. The currently melt-dominated regimes are expected to change to regimes with less discharge in summer and more discharge in winter. In all other regions, at least one model deviates from the majority regime prediction and the direction of change is less clear.

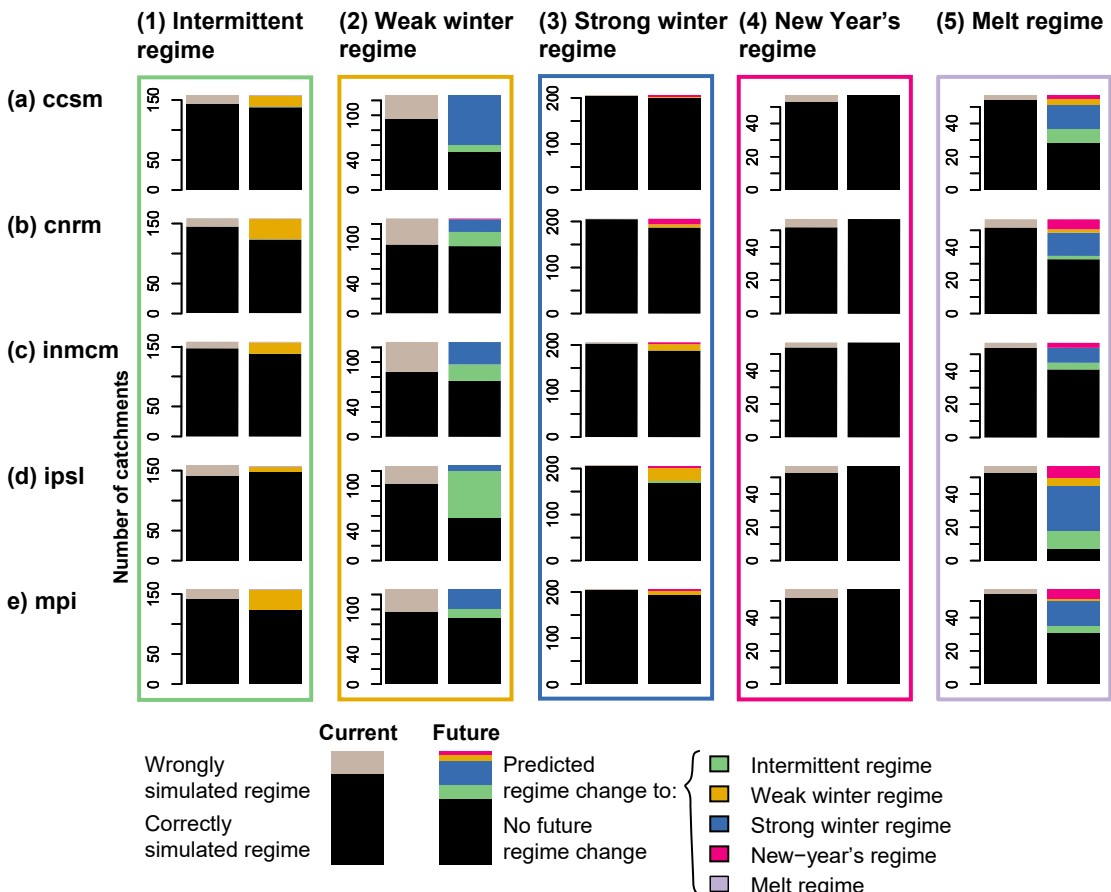

**Figure 6.** Current regime simulation error and future predicted regime changes for the five regimes: (1) Intermittent, (2) weak winter regime, (3) strong winter regime, (4) New Year's regime, and (5) melt regime and the five GCMs: (a) ccsm, (b) cnrm, (c) inmcm, (d) ipsl, and (e) mpi. The number of catchments where the reference simulations result in the observed regime class and a wrong regime class are given in black and grey, respectively. The number of catchments with no predicted regime changes is given in black, the direction of change for the catchments with predicted changes is indicated by the respective regime color.





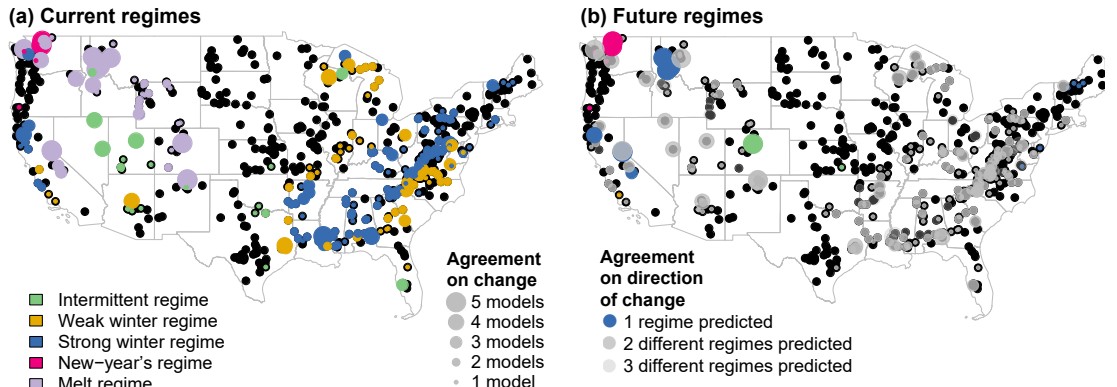

**Figure 7.** (a) Current regimes and agreement of models regarding regime changes. Catchments colored according to their observed regime show catchments where at least one out of the five GCMs predicts a regime class change. The size of the dot indicates the strength of model agreement. (b) Future regimes and agreement of models regarding the direction of change. The size of the dot indicates agreement on change, the color of the dot the agreement on direction of change. All GCMs predict the same change in colored catchments, GCMs disagree on the direction of change in grey catchments where the shading indicates the strength of agreement. Black catchments are either predicted to experience no changes or their reference regime was incorrectly predicted by more than two GCMs.

The future regime changes detected are relatively robust for currently melt-influenced regimes while they are not consistent
for the other regime types. The predicted changes in melt influenced regimes are in line with findings by Coopersmith et al. (2014) who found that snow pack has diminished in the Rocky Mountains in the past and are consistent with future predicted increases in temperature (Vose et al., 2017) and related decreases in snowpack (Easterling et al., 2017). In contrast, predicted changes in precipitation are variable in space and time (Easterling et al., 2017), which disables clear change assessments for rainfall-dominated regimes. Similarly, Milner et al. (2017) and Adam et al. (2009) found on a global scale that warming is
generally associated with reductions in glaciermelt and losses of snowpack, respectively, and therefore changes in streamflow seasonality. However, Adam et al. (2009) also point out that catchments more sensitive to changes in precipitation than temperature may show different change patterns. While our study focused on detecting changes between existing regime classes, there might emerge new regimes (Leng et al., 2016), which we have not considered here.

The changes of melt-influenced towards more rainfall-influenced regimes in the Rocky Mountains and the dependence of
flood and drought timing on the streamflow regime allows us to think about the impacts of regime changes on future extremes. A shift from a melt regime to one of the rainfall-influenced regimes implies a shift of the flood and drought seasons. Under a melt regime, floods mainly occur in spring and early summer when snowmelt and rain-snow interactions enhance the flood signal. In contrast, droughts are mainly observed in winter due to snow accumulation temporally storing water in the catchment. A decreased influence of snow therefore moves the flood season away from spring/early summer into the season with the biggest
precipitation input, which is often winter or spring. Analogously, the drought season moves away from winter into summer and fall, the seasons with the largest precipitation deficits. At the same time, drought and flood magnitudes may also be impacted,





however, the direction of change is less clear there. These expected changes in flood and drought timing and magnitude have potential implications for the predictability of extremes and the spatial coherence in flood and drought occurrence.

## 4 Conclusions

The aim of this study was to (1) develop a flow regime classification scheme beneficial for climate impact assessments and to (2) use this scheme to evaluate future regime changes. We find that the functional clustering approach applied to classify flow regimes is efficient because it uses the temporal information stored in hydrographs thereby sidestepping the computation of a (large) set of streamflow indices and allows for the identification of contiguous regions with similar streamflow regimes. We conclude that the regime behaviour of the 671 US catchments analyzed here can be summarized by five streamflow regime

classes: intermittent regime, weak winter regime, strong winter regime, New Year's regime, and melt regime. These classes are not only similar in their regimes but also their physiographical and meteorological characteristics as well as their extreme streamflow behaviour including the timing and magnitude of droughts and floods. Because of these similarities, we deem the regime classes developed in this study beneficial not only for climate impact assessments but also for model validation and development, the improvement of predictions in ungauged basins, and estimating hydrological model parameters.

Our change impact assessment shows that predicted regime changes are robust in only very few catchments due to model disagreement regarding change and its direction. These GCM-introduced uncertainties demonstrate that predicted regime shifts should be evaluated carefully. Independent of the climate model, however, there is a relatively robust change signal for currently melt-influenced regimes in mountainous catchments even though models do not necessarily agree on the direction of change. Such mountainous catchments take an important role as water towers providing essential freshwater resources to downstream

regions (Immerzeel et al., 2020; Viviroli et al., 2007). Expected changes in these mountainous regions, which are crucial for water supply, point out the potential need for adaptations of water management strategies. Water may need to be stored in reservoirs during winter in order to sustain current summer flows in dependent downstream catchments (Brunner et al., 2019a). A careful evaluation of future regime shifts and their uncertainty can guide decision making on water management and attempt to mitigate the negative impacts of climate change.

*Data availability.* The regime classes derived for the 671 catchments in the CAMELS data set are provided via HydroShare (Brunner, 2020).

*Author contributions.* MIB developed the concept of the study together with MPC. LAM provided the streamflow simulations and model evaluation statistics. MIB established the regime clusters, performed the climate impact assessment on regimes, and wrote the first draft of the manuscript. AJN and AWW provided interpretations of model performance, and AWW contributed the model sensitivity evaluation concept. All co-authors have revised and edited the manuscript.



*Competing interests.* The authors declare that they have no conflict of interest.

*Acknowledgements.* The daily discharge time series used in this study are available via the USGS website: https://waterdata.usgs.gov/nwis
and the CAMELS catchment attributes can be downloaded via https://ral.ucar.edu/solutions/products/camels. This work was supported by the
Swiss National Science Foundation via a PostDoc.Mobility grant (Number: P400P2_183844, granted to MIB), and the National Center for
Atmospheric Research, which is a major facility sponsored by the National Science Foundation under Cooperative Agreement No. 1852977.
AJN and AWW were also partially supported by the US Army Corps of Engineers Climate Preparedness and Resilience Program.





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
