# Peer review of "Future streamflow regime changes in the United States: assessment using functional classification"

_Hydrology and Earth System Sciences, 2020_

## Short Comment (SC1) · 23 Mar 2020

This paper presents an interesting analysis of streamflow and its changes across the USA. The paper i) classifies catchments based on their existing flow regime, and ii) assesses how these different flow regimes are expected to change into the future. Overall this paper seems like an interesting and relevant contribution to HESS, and I enjoyed reading the paper. This short comment is not intended as a full review of the paper, but I hope that sharing the below thoughts may help to strengthen the paper.

\*\*I am aware that there is some degree of (what may be classified as) self-advertising in this comment, but my comments can be addressed without (again) citing the single

[Figure]

self-reference that I provide.**

A key assertion and motivation of this study is that hydrological classifications have not really incorporated "temporal information in clustering hydrological catchments, [. . .] even though such information is potentially very useful". This statement is used as a motivation to develop a classification that incorporates such information.

Overall, this is a good idea. However, while many studies indeed ignore the temporal aspect, there are existing studies that explicitly incorporate this information into their classifications (leading to very similar types of classifications as presented in this paper).

Grouping based on seasonal regimes have been introduced a long time ago: e.g.:

Pardé, M. (1960). The river regime in New Zealand. Revue de Géographie Alpine , 48 (3), 383-429. (And probably also in earlier works of Pardé and others).

and have been applied globally: Haines, A. T., Finlayson, B. L., & McMahon, T. A. (1988). A global classification of river regimes. Applied Geography, 8(4), 255–272. https://doi. org/10.1016/0143-6228(88)90035-5,

which classified global streamflow regimes into very similar type classes as done in the presented HESSD manuscript. (However, obviously with a greater variety of classes since the global spectrum of river flows was taken into account).

A global analysis has been updated: Knoben, W. J., Woods, R. A., & Freer, J. E. (2018). A quantitative hydrological climate classification evaluated with independent streamflow data. Water Resources Research, 54(7), 5088-5109.

This Knoben study also includes temporal information of streamflow classes into the final classification, which again is very similar in nature to what is presented in the presented HESSD manuscript. Different to the presented manuscript is that the classification metrics are not based on streamflow themselves directly. However, the classes it produces are shown to have very similar within-class seasonal streamflow regimes,

which seems to make them functionally equivalent to what is presented in the HESSD paper.

In addition, such analyses are also available for the United States, focusing on seasonal streamflow regimes: Coopersmith, E., Yaeger, M. A., Ye, S., Cheng, L., & Sivapalan, M. (2012). Exploring the physical controls of regional patterns of flow duration curves-Part 3: A catchment classification system based on regime curve indicators. Hydrology and Earth System Sciences, 16(11), 4467.

and seasonal streamflow (and all other water components) regimes: Berghuijs, W. R., Sivapalan, M., Woods, R. A., & Savenije, H. H. G. (2014). Patterns of similarity of seasonal water balances: A window into streamflow variability over a range of time scales. Water Resources Research, 50, 5638–5661. (whereby this study, based on largely similar classes, also had similar conclusions regarding class correlations with e.g. aridity, snowiness, flood timing, low flow timing.)

I understand that these studies have already mostly been cited in the main text, but their validity as classifications of seasonal flow regimes that include temporal information has sort of been dismissed by the statement that "The use of catchment characteristics can be problematic because there is often no clear link between these characteristics and streamflow indices". Yet, all of the above-listed studies (except Pardé maybe) show explicitly how their classifications lead to similar within-class behavior of seasonal streamflow regimes.

I think there is an opportunity to slightly reframe the paper to acknowledge that this study complements existing classifications that also incorporate temporal information of flow regimes, (rather than to imply that nothing (useful) exists in this field). (Or alternatively, be more precise and explicit about what the previous classifications can't do that yours does).

The use of B-spline basis functions to characterize the streamflow regimes functional behavior in this HESS manuscript seems to be a useful addition to existing literature

that I look forward to seeing published in HESS.

---

## Referee Comment (RC1) · Anonymous Referee #1 · 30 Mar 2020

General comments: This paper develops a functional hydrologic classification method, applies it to a large number of gages in the continental US, and assesses future changes in hydrologic regime resulting from climate change using VIC model output driven by downscaled GCMs. The paper makes an important contribution to the literature on hydrologic classification because its use of functional data analysis, in which each hydrograph is represented as a function, addresses issues associated both with top-down classifications based on catchment characteristics (lack of process connections between those characteristics and streamflow) and with bottom-up classifications based on streamflow indices (requiring selection of indices that may not fully describe the hydrologic regime). As such, the paper provides an interesting and innovative

approach that is useful for assessing hydrologic regimes for climate-change impact assessment or to predict the behavior of ungaged catchments.

Specific comments: Line 30: It should be noted upfront that hydrologic regimes, defined here as those "described by mean annual hydrographs", can encompass a broader set of variables than those used here. In particular, the functional approach that analyzes mean annual hydrographs describes the seasonal patterns of stream-flow, but not patterns that occur on either shorter timescales (e.g., flashiness) or longer timescales (e.g., interannual variability). For analyzing these types of changes in hy-drologic regime, the functional analysis is less useful and particular streamflow indices must instead be used. This is important because the seasonal hydrologic regime is highly sensitive to changes in temperature in the melt region because of the snow sig-nal, but as noted later in the paper (lines 291-292), there is no clear seasonal signal in catchments that are more sensitive to changes in precipitation to temperature. It is not that those catchments will not experience hydrologic changes due to climate change, it is just that the type of changes they experience (e.g., greater flashiness or interan-nual variability) are not captured by the functional analysis and its focus on seasonal changes. Also, although at the seasonal scale of analysis meteorological variables were more significant than physiographical variables in predicting class membership (lines 228-229), it is possible that physiographical characteristics would be more sig-nificant in determining class membership when it comes to flashiness or interannual variability, because of the role of land-surface characteristics like lithology, soil, and vegetation in mediating the climatic signal.

Lines 233-234: Would it be possible to include a map of nominal hydrologic class for ungaged catchments? If the random forest model can predict hydrologic class based on meteorological and physiographical variables, it should be able to apply the classi-fication and predict hydrologic class for all the catchments in the CONUS (assuming data for the predictor variables are available CONUS-wide). That would be an interest-ing map to see because it would further illustrate the spatial contiguity and extent of

the hydrologic classes beyond the gaged catchments in Figure 2.

Technical corrections: Line 243: Typo: "Klomogorov" should be "Kolmogorov". Line 298: Is "temporally" meant to be "temporarily"? That makes more sense to me.

---

## Referee Comment (RC2) · Florian Ulrich Jehn (Referee) · 15 Apr 2020

General evaluation: This paper proposes a new method to cluster catchment based on the temporal information in their hydrological regime and uses the found regime clusters to evaluate how climate change will change the regimes clusters. I think this is an interesting approach and yields good results, especially the changing of the regimes due to climate change. In addition, the paper is overall well written and has a good flow to it. I think it can be published after minor revisions. However, I have one larger points where I think clarification is necessary.

Main point:

- Line 98: Why those five spline basis functions? How can you be sure that those are enough to represent diverse regimes? It there a connection between using five spline basis function and finding five streamflow regimes? I think this part should be extended to make it clearer what those decisions where based upon.

Minor Points:

- Line 47: "The use of catchment characteristics can be problematic because there is often no clear link between these characteristics and streamflow indices (Ali et al., 2012; Addor et al., 2018)." I think this is worded a bit too strict. For example, Addor et al. 2018 indeed showed that there are differences between the link of catchment characteristics and streamflow indices, but they also showed this connection can be relatively strong for some catchment attributes. Overall, I think this section should be less dismissive of the findings of the cited papers.

- Figure 2 and its discussion: As you are already citing my paper, I hope it is appropriate to mention that that the final version is now published in HESS (https://www.hydrolearth-syst-sci.net/24/1081/2020/hess-24-1081-2020.html) and discusses the different flow regimes in CAMELS in more depth than before (Figure 6). I think the results are very similar to the ones in this paper, but also show that the flow regimes found here can be split in more distinct groups. However, this might be more of a question of the desired granularity.

- Line 85: How is satisfactory model performance defined?

- Line 90 and following, code availability: I did not see any link to a code repository for this paper (my apologies if I missed it). I think in a paper that does propose a new method, it is important to provide the code used. While the method section explains the idea well, it is still a non-trivial task to recreate the method of this paper without any code examples to work with.

- Line 226: I am unsure if avoiding small clusters is a sign of a good clus-

HESSD
tering. As river behavior is a natural process, I would expect it to follow some kind of normal distribution, which would result in some bigger clusters and some smaller, more extreme clusters. For future research, it might be a good idea to explore a continuous classification as done by Knoben et al (https://agupubs.onlinelibrary.wiley.com/doi/abs/10.1029/2018WR022913) or a fuzzy clustering approach to avoid the arbitrary cut-off points of clusters.

- Line 256: The difference here might again be a question of granularity. Especially the catchments in Florida behave very uniquely.

- Figure 5: It is very difficult to distinguish the lines from each other here. I think it might be a good idea to increase the size of this figure. Also, I would recommend to use more easily distinguishable colors.

Kind regards, Florian Ulrich Jehn

---

## Author Comment (AC1) · 23 Apr 2020

**Commentator: Wouter Berghuijs**

This paper presents an interesting analysis of streamflow and its changes across the USA. The paper i) classifies catchments based on their existing flow regime, and ii) assesses how these different flow regimes are expected to change into the future. Overall this paper seems like an interesting and relevant contribution to HESS, and enjoyed reading the paper. This short comment is not intended as a full review of the paper, but I hope that sharing the below thoughts may help to strengthen the paper.**I am aware that there is some degree of (what may be classified as) self-advertising in this comment, but my comments can be addressed without (again) citing the single self-reference that I provide.**
A key assertion and motivation of this study is that hydrological classifications have not really incorporated "temporal information in clustering hydrological catchments, [...] even though such information is potentially very useful". This statement is used as a motivation to develop a classification that incorporates such information. Overall, this is a good idea. However, while many studies indeed ignore the temporal aspect, there are existing studies that explicitly incorporate this information into their classifications (leading to very similar types of classifications as presented in this paper). Grouping based on seasonal regimes have been introduced a long time ago: e.g.:
Pardé, M. (1960). The river regime in New Zealand. Revue de Géographie Alpine , 48(3), 383-429. (And probably also in earlier works of Pardé and others) and have been applied globally:
Haines, A. T., Finlayson, B. L., & McMahon, T. A.(1988). A global classification of river regimes. Applied Geography, 8(4), 255–272.https://doi. org/10.1016/0143-6228(88)90035-5, which classified global streamflow regimes into very similar type classes as done in the presented HESSD manuscript. (However, obviously with a greater variety of classes since the global spectrum of river flows was taken into account). A global analysis has been updated: Knoben, W. J., Woods, R. A., & Freer, J. E.(2018). A quantitative hydrological climate classification evaluated with independent streamflow data. Water Resources Research, 54(7), 5088-5109. This Knoben study also includes temporal information of streamflow classes into the final classification, which again is very similar in nature to what is presented in the presented HESSD manuscript. Different to the presented manuscript is that the classification metrics are not based on streamflow themselves directly. However, the classes it produces are shown to have very similar within-class seasonal streamflow regimes, which seems to make them functionally equivalent to what is presented in the HESSD paper. In addition, such analyses are also available for the United States, focusing on seasonal streamflow regimes: Coopersmith, E., Yaeger, M. A., Ye, S., Cheng, L., & Sivapalan, M.(2012). Exploring the physical controls of regional patterns of flow duration curves-Part3: A catchment classification system based on regime curve indicators. Hydrology and Earth System Sciences, 16(11), 4467.and seasonal streamflow (and all other water components) regimes: Berghuijs, W. R.,Sivapalan, M., Woods, R. A., & Savenije, H. H. G. (2014). Patterns of similarity of seasonal water balances: A window into streamflow variability over a range of timescales. Water Resources Research, 50, 5638–5661. (whereby this study, based on largely similar classes, also had similar conclusions regarding class correlations with e.g. aridity, snowiness, flood timing, low flow timing.) I understand that these studies have already mostly been cited in the main text, but their validity as classifications of seasonal flow regimes that include temporal information has sort of been dismissed by the statement that "The use of catchment characteristics can be problematic because there is often no clear link between these characteristics and streamflow indices". Yet, all of the above-listed studies (except Pardé maybe) show explicitly how their classifications lead to similar within-class behavior of seasonal streamflow regimes. I think there is an opportunity to slightly reframe the paper to acknowledge that this study complements existing classifications that also incorporate temporal information

of flow regimes, (rather than to imply that nothing (useful) exists in this field). (Or alternatively, be more precise and explicit about what the previous classifications can't do that yours does). The use of B-spline basis functions to characterize the streamflow regimes functional behavior in this HESS manuscript seems to be a useful addition to existing literature that I look forward to seeing published in HESS.

**Reply:** *Dear Wouter, thank you for your thoughts on the framing of our manuscript, which we considered while revising our manuscript. We did by no means intend to imply that nobody has ever looked at timing related streamflow indices in clustering but rather wanted to point out that studies that explicitly consider the temporal information in the continuous signal are rare. This is why we wrote: 'Both the catchment and climate characteristics and the streamflow index approaches neglect* nearly *all available temporal information embedded in a streamflow time series or regime* in the form of temporal (auto-) correlation'. *The keywords here are 'nearly' and 'in the form of temporal (auto-) correlation'. We agree that the subsequent sentence may seem exclusive of some contributions and rephrased it to explicitly state that some studies have clustered on streamflow indices related to seasonality and timing: 'While some of the index-based approaches have considered indices related to streamflow timing and seasonality [Haines et al., 1988; Bower et al., 2004; McCabe and Wolock, 2014], only very few studies have tried to explicitly take account of temporal streamflow information in clustering hydrological catchments, e.g. by using the shape of the autocorrelation function as an index [Toth, 2013], even though such information is potentially very useful.' We also rephrased the sentence on the weak link between certain streamflow and catchment characteristics to: 'The use of catchment characteristics is not always beneficial as certain streamflow indices do not show clear links to these characteristics [Ali et al., 2012; Addor et al., 2018].' We cite the Knoben et al. [2018] study under clustering approaches related to climate characteristics as their 'classification scheme is based only on climatic information and can be evaluated with independent streamflow data.' We acknowledge the work by Coopersmith et al. [2014] under approaches using streamflow and climate characteristics as they use date of maximum runoff as a streamflow index, which is related to time but does not say anything about the changes of streamflow over time. We acknowledge the work by Berghuijs et al. [2014] under the climate indices clustering approaches (formerly mixed approaches). You use a measure for the strength of precipitation seasonality but do not directly include information on the temporal distribution of precipitation over the season. We hope that you find the updated framing more precise and inclusive.*

**References used in this response**

Addor, N., G. Nearing, C. Prieto, A. J. Newman, N. Le Vine, and M. P. Clark (2018), A ranking of hydrological signatures based on their predictability in space, *Water Resour. Res.*, *54*, 8792–8812, doi:10.1029/2018WR022606.

Ali, G., D. Tetzlaff, C. Soulsby, J. J. McDonnell, and R. Capell (2012), A comparison of similarity indices for catchment classification using a cross-regional dataset, *Adv. Water Resour.*, *40*, 11–22, doi:10.1016/j.advwatres.2012.01.008.

Berghuijs, W. R., M. Sivapalan, R. A. Woods, and H. H. G. Savenije (2014), Patterns of similiarity of seasonal water balances: A window into streamflow variability over a range of time secales, *Water Resour. Res.*, *50*, 5638–5661, doi:10.1002/2014WR015692.

Bower, D., D. M. Hannah, and G. R. McGregor (2004), Techniques for assessing the climatic sensitivity of river flow regimes, *Hydrol. Process.*, *18*(13), 2515–2543,

doi:10.1002/hyp.1479.

Coopersmith, E. J., B. S. Minsker, and M. Sivapalan (2014), Patterns of regional hydroclimatic shifts: An analysis of changing hydrologic regimes, *Water Resoures Researh*, *50*, 1960–1983, doi:10.1111/j.1752-1688.1969.tb04897.x.

Haines, A. T., B. L. Finlayson, and T. A. McMahon (1988), A global classification of river regimes, *Appl. Geogr.*, *8*(4), 255–272, doi:10.1016/0143-6228(88)90035-5.

Knoben, W. J. M., R. A. Woods, and J. E. Freer (2018), A quantitative hydrological climate classification evaluated with independent streamflow data, *Water Resour. Res.*, *54*(7), 5088–5109, doi:10.1029/2018WR022913.

McCabe, G. J., and D. M. Wolock (2014), Spatial and temporal patterns in conterminous United States streamflow characteristics, *Geophys. Res. Lett.*, *41*(19), 6889–6897, doi:10.1002/2014GL061980.

Toth, E. (2013), Catchment classification based on characterisation of streamflow and precipitation time series, *Hydrol. Earth Syst. Sci.*, *17*(3), 1149–1159, doi:10.5194/hess-17-1149-2013.

---

## Author Comment (AC2) · 23 Apr 2020

**Reviewer 1**

**General comments:**

This paper develops a functional hydrologic classification method, applies it to a large number of gages in the continental US, and assesses future changes in hydrologic regime resulting from climate change using VIC model output driven by downscaled GCMs. The paper makes an important contribution to the literature on hydrologic classification because its use of functional data analysis, in which each hydrograph is represented as a function, addresses issues associated both with top-down classifications based on catchment characteristics (lack of process connections between those characteristics and streamflow) and with bottom-up classifications based on streamflow indices (requiring selection of indices that may not fully describe the hydrologic regime). As such, the paper provides an interesting and innovative approach that is useful for assessing hydrologic regimes for climate-change impact assessment or to predict the behavior of ungaged catchments.

**Specific comments:**

Line 30: It should be noted upfront that hydrologic regimes, defined here as those "described by mean annual hydrographs", can encompass a broader set of variables than those used here. In particular, the functional approach that analyzes mean annual hydrographs describes the seasonal patterns of stream-flow, but not patterns that occur on either shorter timescales (e.g., flashiness) or longer timescales (e.g., interannual variability). For analyzing these types of changes in hydrologic regime, the functional analysis is less useful and particular streamflow indices must instead be used. This is important because the seasonal hydrologic regime is highly sensitive to changes in temperature in the melt region because of the snow signal, but as noted later in the paper (lines 291-292), there is no clear seasonal signal in catchments that are more sensitive to changes in precipitation to temperature. It is not that those catchments will not experience hydrologic changes due to climate change, it is just that the type of changes they experience (e.g., greater flashiness or inter annual variability) are not captured by the functional analysis and its focus on seasonal changes. Also, although at the seasonal scale of analysis meteorological variables were more significant than physiographical variables in predicting class membership (lines 228-229), it is possible that physiographical characteristics would be more significant in determining class membership when it comes to flashiness or interannual variability, because of the role of land-surface characteristics like lithology, soil, and vegetation in mediating the climatic signal.

**Reply:** *Thank you for highlighting this point. We agree that the functional clustering approach presented here does not consider similarities in inter-annual variability when clustering catchments, also not in the change assessment. While the approach does indeed not allow for the consideration of flashiness at an event scale, it allows for a partial consideration of flashiness as the mean annual hydrographs used for the clustering and the change analysis have a daily temporal resolution. We make the following addition when comparing the FDA to the index-based clustering approaches:*

*This scheme makes better use of the seasonal and temporal information stored in the hydrological regime than index-based approaches. However, it does neither consider streamflow patterns at short, event time scales such as flashiness, nor time scales longer than a year as for instance caused by inter-annual variability.*

*We acknowledge in the results section that: 'Catchments without predicted regime changes may still undergo changes in specific streamflow characteristics, such as variability or low and high flows (l.274-275).'*

*We differentiate the discussion of class predictor strengths by adding: 'The relationship of class membership to physiographical characteristics may be weaker than the one to climatic*

*characteristics as the clusters are formed using the mean annual hydrographs whose seasonality is strongly influenced by climate. The link to physiographical characteristics may be stronger if streamflow characteristics at an event time scale are considered.'*

Lines 233-234: Would it be possible to include a map of nominal hydrologic class for ungaged catchments? If the random forest model can predict hydrologic class based on meteorological and physiographical variables, it should be able to apply the classification and predict hydrologic class for all the catchments in the CONUS (assuming data for the predictor variables are available CONUS-wide). That would be an interesting map to see because it would further illustrate the spatial contiguity and extent of the hydrologic classes beyond the gaged catchments in Figure 2.
**Reply:** *We agree that it would be nice to produce a map of predicted regime classes over the whole CONUS. We did not do this for two main reasons: First, the dataset of physiographical variables we used for this analysis is limited to the 671 catchments in the CAMELS dataset [Newman et al., 2015; Addor et al., 2017]. Furthermore, our classification is limited to streamflow regimes resulting from natural conditions. A classification for all catchments in the CONUS would need to encompass classes for 'human-influenced' catchments where the streamflow regime has been (strongly) altered by water abstractions and transfers, reservoir operation or other human interventions.*

**Technical corrections:**
Line 243: Typo: "Klomogorov" should be "Kolmogorov".
**Reply:** *We corrected this typo.*

Line298: Is "temporally" meant to be "temporarily"? That makes more sense to me
**Reply:** *Yes, we changed temporally to temporarily.*

**References used in this response to the reviewer**

Addor, N., A. J. Newman, N. Mizukami, and M. P. Clark (2017), The CAMELS data set: Catchment attributes and meteorology for large-sample studies, *Hydrol. Earth Syst. Sci.*, *21*(10), 5293–5313, doi:10.5194/hess-21-5293-2017.

Newman, A. J. et al. (2015), Development of a large-sample watershed-scale hydrometeorological data set for the contiguous USA: Data set characteristics and assessment of regional variability in hydrologic model performance, *Hydrol. Earth Syst. Sci.*, *19*(1), 209–223, doi:10.5194/hess-19-209-2015.

---

## Author Comment (AC3) · 23 Apr 2020

**Reviewer 2 : Florian Ulrich Jehn**

**General evaluation:**
This paper proposes a new method to cluster catchments based on the temporal information in their hydrological regime and uses the found regime clusters to evaluate how climate change will change the regimes clusters. I think this is an interesting approach and yields good results, especially the changing of the regimes due to climate change. In addition, the paper is overall well written and has a good flow to it. I think it can be published after minor revisions. However, I have one larger points where I think clarification is necessary.

**Main point:**
- Line 98: Why those five spline basis functions? How can you be sure that those are enough to represent diverse regimes? It there a connection between using five spline basis function and finding five streamflow regimes? I think this part should be extended to make it clearer what those decisions where based upon.
**Reply:** *Thank you for expressing your concern regarding a suitable choice of spline basis functions. We chose five spline basis functions because our tests showed that a further increase in the number of spline bases did not further improve the clustering results. This is confirmed by the overall silhouette width, which is for more spline basis functions (6 to 10) lower or very similar to the one for five basis functions (at five clusters). There is no relation between the number of spline bases and the number of regimes chosen. A choice of four or five clusters would also be optimal if more than five spline bases were used.*
*We added the following specification to the text: 'The suitability of five spline basis functions is confirmed by the overall silhouette width, which is for more spline basis functions (6 to 10) lower or very similar to the one for five basis functions.'*

**Minor Points:**
- Line 47: "The use of catchment characteristics can be problematic because there is often no clear link between these characteristics and streamflow indices (Ali et al., 2012; Addor et al., 2018)." I think this is worded a bit too strict. For example, Addor et al. 2018 indeed showed that there are differences between the link of catchment characteristics and streamflow indices, but they also showed this connection can be relatively strong for some catchment attributes. Overall, I think this section should be less dismissive of the findings of the cited papers.
**Reply:** *We did by no means intend to be dismissive of findings of other papers as we appreciate their work. Furthermore, we do not think that it is a bad thing to find weak relationships between certain streamflow and catchment characteristics. We rephrased the sentence to: 'The use of catchment characteristics is not always beneficial as certain streamflow indices do not show clear links to these characteristics [Ali et al., 2012; Addor et al., 2018]'. Furthermore, based on the point raised by the other reviewer, we stress that the relation between catchment characteristics and streamflow might be more apparent for event-based signatures.*

- Figure 2 and its discussion: As you are already citing my paper, I hope it is appropriate to mention that that the final version is now published in HESS (https://www.hydrolearth-syst-sci.net/24/1081/2020/hess-24-1081-2020.html) and discusses the different flow regimes in CAMELS in more depth than before (Figure 6). I think the results are very similar to the ones in this paper, but also show that the flow regimes found here can be split in more distinct groups. However, this might be more of a question of the desired granularity.
**Reply:** *Thank you for pointing out the necessity to update the reference. We point out the*

*similarity between the clusters resulting from your and our analysis by saying: 'The five regime clusters identified also show spatial similarities with the ten catchment clusters formed by (Jehn2019) for the same set of catchments using a small set of hydrological streamflow characteristics.' (l.225-226). For some applications it may indeed be useful to have more distinctive clusters.*

- Line 85: How is satisfactory model performance defined?

**Reply:** *Melsen et al.* (2018) *who produced the simulated streamflow data used in this study provide simulations only for the subset of 605 catchments. They chose to run simulations for this subset only as different data sources disagreed on catchment size for the remaining catchments. A reliable estimate of catchment size is crucial to ensure accurate forcing input for the lumped model. We reformulated the sentence as follows: 'In contrast, the regime change analysis uses streamflow simulated by the hydrological Variable Infiltration Capacity (VIC) model for a subset of 605 catchments, for which reliable data on catchment area was available at the time the simulations were produced [Melsen et al., 2018]. Kling-Gupta efficiencies obtained over these basins with VIC varied from a first quartile of 0.47, a median of 0.6 and a third quartile of 0.71, with the lowest values obtained in the Great Plains'. (l.145, p.6)*

- Line 90 and following, code availability: I did not see any link to a code repository for this paper (my apologies if I missed it). I think in a paper that does propose a new method, it is important to provide the code used. While the method section explains the idea well, it is still a non-trivial task to recreate the method of this paper without any code examples to work with.

**Reply:** *It is correct that we have not yet provided a link to the repository with the catchment clusters. This is because we wanted to wait for the DOI of this manuscript to be available before we created a DOI for the dataset. The dataset can now be accessed via HydroShare: The link to the dataset was added to the manuscript (https://doi.org/10.4211/hs.069f552f96ef4e638f4bec281c5016ad).*
*To facilitate the reproduction of the functional data clustering approach, we added details on the R-packages and functions we used: 'The analysis is performed in R using the packages fda.usc [Febrero-Bande and Oviedo de la Fuente, 2012] and fda [Ramsay et al., 2014] and the following functions: (1) conversion of regimes to functional data objects: fdata, (2) creating of B-spline basis functions: create.bspline.basis, (3) computation of spline coefficients for all regimes: Data2fd. The clustering into regime classes is performed using the R-package stats (R Core Team, 2019). A Euclidean distance matrix is computed using the matrix of n= 671×5 spline coefficients (Figure 1.2 a–b) (dist). We use a hierarchical clustering algorithm (hclust) allowing for non-elliptical clusters (Gordon, 1999) with Ward's minimum variance criterion, which minimizes the total within-cluster variance (Ward, 1963). To identify an optimal number of clusters, we cut the tree a tk= 2,...,30 clusters (cutree) and compute the mean silhouette width (Rousseeuw, 1987), which provides a measure of clustering validity, for the different numbers of clusters'*

- Line 226: I am unsure if avoiding small clusters is a sign of a good clustering. As river behavior is a natural process, I would expect it to follow some kind of normal distribution, which would result in some bigger clusters and some smaller, more extreme clusters. For future research, it might be a good idea to explore a continuous classification as done by Knoben et al (https://agupubs.onlinelibrary.wiley.com/doi/abs/10.1029/2018WR022913) or a fuzzy clustering approach to avoid the arbitrary cut-off points of clusters.

**Reply:** *One can always form new clusters of 'special cases' to decrease the within cluster variability. However, the formation of more clusters in our case resulted in a decrease in average silhouette width, which is not desirable. Furthermore, the formation of more clusters with smaller*

*in-between cluster differences, would have diverted the focus from the detection of major regime changes to the detection of minor regime changes. We agree that the use of probabilistic instead of deterministic class memberships may be desirable for some applications. For this particular application, however, we found deterministic clusters to be more appropriate.*

- Line 256: The difference here might again be a question of granularity. Especially the catchments in Florida behave very uniquely.
**Reply:** *The finding that the correct regime class is not simulated in certain regions when forcing the model with GCM output is likely related to the fact that certain processes in these areas are not well represented by GCMs.*

- Figure 5: It is very difficult to distinguish the lines from each other here. I think it might be a good idea to increase the size of this figure. Also, I would recommend to use more easily distinguishable colors.
**Reply:** *Thank you for pointing out the need to increase the legibility of this figure. We rearranged the subplots of this figure in order to increase the size of the individual subplots and darkened the color of the control regime to increase contrast with respect to the regimes simulated using the GCM output.*
**Modification: Figure 5**

**References used in this response to the reviewers**

Addor, N., G. Nearing, C. Prieto, A. J. Newman, N. Le Vine, and M. P. Clark (2018), A ranking of hydrological signatures based on their predictability in space, *Water Resour. Res.*, *54*, 8792–8812, doi:10.1029/2018WR022606.

Ali, G., D. Tetzlaff, C. Soulsby, J. J. McDonnell, and R. Capell (2012), A comparison of similarity indices for catchment classification using a cross-regional dataset, *Adv. Water Resour.*, *40*, 11–22, doi:10.1016/j.advwatres.2012.01.008.

Febrero-Bande, M., and M. Oviedo de la Fuente (2012), Statistical Computing in Functional Data Analysis: The R Package fda.usc, *J. Stat. Softw.*, *51*(4), 1-3-, doi:10.18637/jss.v051.i04.

Melsen, L., N. Addor, N. Mizukami, A. Newman, P. Torfs, M. Clark, R. Uijlenhoet, and R. Teuling (2018), Mapping (dis) agreement in hydrologic projections, *Hydrol. Earth Syst. Sci.*, *22*, 1775–1791, doi:10.5194/hess-22-1775-2018.

Ramsay, J. O., H. Wickham, S. Graves, and G. Hooker (2014), Package "fda": Functional data analysis, *CRAN*.

---

## Referee Comment (RC3) · Genevieve Ali (Referee) · 5 May 2020

**GENERAL COMMENTS**

In this manuscript entitled "Future streamflow regime changes in the United States: assessment using functional classification", two main goals are pursued: (1) develop a catchment classification scheme for streamflow regimes, and (2) use this scheme to evaluate changes in future flow regimes. Contrary to the majority of previously published catchment classification efforts, here the authors decided not to rely on streamflow indices. Instead, they are using a functional approach via which the shapes of mean annual hydrographs are classified, this in order to retain temporal autocorrelation information. Overall, the manuscript is of appropriate length, well written and with good-quality figures and tables. The dual focus of the manuscript on catchment classification and climate change impact assessment is very interesting, and I agree with the authors about their description of the advantages of functional classification. I did find that a few statements made in the manuscript warranted clarification, and that some details regarding the datasets, process interpretations, or linkages with existing literature were lacking (see specific comments below). With revisions, I believe that this manuscript will be interesting to the HESS readership, and a great addition to our growing body of literature on catchment classification.

**

SPECIFIC COMMENTS

N.B.: page and line numbers are noted as PX (page X) and LX (line X).

Section 2.1: Given the international readership of HESS, I think that more detailed information is needed about the catchment selection criteria. For people not familiar with the CAMELS dataset, it is quite unclear what is meant by "minimum human impact": is the human impact assessed in terms of catchment-wide land use (that would mean no agricultural or urban catchment), or river regulation? And how may the answer to that question affect the generalization potential of the manuscript conclusions? In other words, the authors should discuss the limitations associated with not considering human-impacted catchments in the present study... Also, how was the 1981-2018 data period chosen for the analysis?

P5 L120: There is a reference to characteristics with missing values. Which characteristics (or types of characteristics) are the ones with missing values? Did omitting them lead to biased results?

Section 2.5, specifically L180-182: How was the comparison made, exactly, from a quantitative or statistical standpoint? Using contingency tables or crosstabs? Or some-

[Figure]

thing else? This is a bit unclear to me. . .. Maybe because I was expecting a statistical comparison when in fact, it is not what was done. . .

Figure 3: The different (graphical) features of the boxplots should probably be described in the figure caption. I assume that the horizontal black lines refer to the medians.... what do the whiskers represent, though: 1 interquartile range (IQR), 1.5 IQR, min and max values, or something else? Are there no statistical outliers associated with each cluster, i.e., each individual box?

P9 L203-204: That should not be a surprise, given that the flood and drought definitions are hydrograph-based.... or am I missing something?

P9-10, L207-210: The text description, here, does not underline that strong of a contrast between the weak winter regime and the strong winter regime. Maybe it can be rephrased for the contrast to be expressed more strongly?

P10 L217-218: That would explain why there is such a large degree of spatial contiguity/spatial autocorrelation within each cluster. However, it is a bit unclear to me, from the text, whether a RF classification using climatological variables only performs equally as well as – or better than – a RF classification that used both climatological and physiographic variables.

P10 227: The authors stated that "However, our clustering scheme avoids the formation of very small clusters seen in Jehn et al. (2019)." First, what might explain this? Second, the authors seem to imply that having very small clusters is an inconvenient, and I am not sure I agree – very small clusters could represent very local conditions or hotspots, which are real. The authors should either rephrase or at least nuance their statement to clarify what they mean.

P10 L230-234: The authors wrote that "The strong link between regime classes and meteorological and physiographical catchment characteristics allows for the attribution of ungauged catchments, where streamflow data are not available, to one of the regime

classes, which is potentially very useful for the prediction of streamflow characteristics in ungauged basins". I am not sure where that statement is coming from, as ungauged catchments were not examined in the present study. I agree that the present study might have interesting implications for predictions in ungauged catchments, but this statement, as written, reads as a result when in fact it is an interpretation.

In the same line of thought, I wonder whether it would be possible to have separate Results and Discussion sections in the manuscript. There are a few instances, in the text, where it can be tricky to distinguish whether a plain result/fact is being stated, or whether a hypothesis/interpretation is being put forward.

Figure 4: This figure is quite interesting but the comparison of "climate sensitivity" between observations and simulations appears quite qualitative. I wonder: 1) How were the five example catchments showcased in this figure chosen (or, are those sites representative of median cluster conditions)?; and 2) Was a quantitative method of comparison between observations and simulations used for all catchments?

P11 L240: The authors refer to a "visual analysis"; were all plots for all 605 catchments visually analyzed?

P11 L243-244: The Methods section should explicitly state what the Kolmogorov-Smirnov test was used for, the assumptions being it, and the null and alternate hypotheses (so that readers know what the test results mean). Also, a test cannot be rejected: we can only reject or fail to reject a null hypothesis, so that sentence should be reworded.

Figure 5: Lines are a bit difficult to distinguish on this figure; making it larger and changing the symbology might help.

P12 L258-259: The authors wrote "In contrast, regimes with a strong seasonality such as strong winter and New Year's regimes are well simulated". What about the melt regime, which is also highly seasonal?

Figure 7: If the black circles mean no regime change, the legend should state so.

\*\*

COMMENTS SPECIFIC TO DISCUSSION ELEMENTS WORTH INCLUDING IN THE MANUSCRIPT

Discussion comment #1: In the present study, regime clusters appear equivalent to clusters derived based on physiographic similarity and clusters derived based on climatological similarity... this is contrary to studies published by Ali et al. (2012) and Oudin et al. (2010) – in a comforting way, I might add – and this should probably be discussed. The "overlap" or agreement between the different classifications bodes well for using climatic and physiographic information as a proxy for streamflow regime types. The fact that an agreement was found in the present study and not in others may be due to the fact that here, functional data were used instead of select streamflow indices.

Discussion comment #2: It is not a study limitation per se, but the authors may want to discuss the rationale for using functional streamflow data classification (to preserve temporal information) while NOT using climate timeseries (e.g., mean annual hyetograph) for classification purposes. When I started reading the manuscript, I was puzzled by the fact that a classification based on temporally autocorrelated data (i.e., whole annual hydrographs) was going to be compared to a classification based on climate indices. In other words, I wondered how the analyses would turn out given that different regions may have similar values of mean annual precipitation, even though the temporal distribution of that precipitation may be skewed in some places but not elsewhere. In the end, the authors found that they could neglect the temporal information included in climate timeseries and still manage to use that climate information (i.e., the climate index class) as a good proxy for streamflow regime class (which, itself, is based on temporally autocorrelated data). That warrants discussion, I think, as it is a bit counter-intuitive (to me, anyway...)

Discussion comment #3: The authors may want to use the concepts of resistance,

resilience and synchronicity discussed by Carey et al. (2010): those concepts partly echo what the authors are referring to as "climate sensitivity".

**

EDITORIAL SUGGESTIONS

P2 L30: "illustrate the hydrological functioning" seems more appropriate than "govern the hydrological functioning", since the authors are referring to streamflow regimes

P2 L31: I think that the phrase "influencing streamflow variability" should be changed.... Otherwise the whole sentence read as "The characteristics of streamflow regimes [influence] streamflow variability and seasonality", which reads as a circular statement.

P10 L217: "shows that the the most important variables for" SHOULD BE CHANGED FOR "shows that the most important variables for"

P11 L243: "Klomogorov–Smirnov" SHOULD BE CHANGED FOR "Kolmogorov-Smirnov"

P13 L274: "In contract" SHOULD BE CHANGED FOR "In contrast"

**

REFERENCES CITED IN THIS REVIEW

Ali, G., Tetzlaff, D., Soulsby, C., McDonnell, J. J., and Capell, R. (2012), A comparison of similarity indices for catchment classification using a cross-regional dataset. Advances in Water Resources, 40, 11-22. doi:10.1016/j.advwatres.2012.01.008

Carey, S.K., Tetzlaff, D., Seibert, J., Soulsby, C., Buttle, J., Laudon, H., McDonnell, J., McGuire, K., Caissie, D., Shanley, J., Kennedy, M., Devito, K. and Pomeroy, J.W. (2010), Inter-comparison of hydro-climatic regimes across northern catchments: synchronicity, resistance and resilience. Hydrological Processes, 24: 3591-

3602. doi:10.1002/hyp.7880

Oudin, L., Kay, A., Andréassian, V., and Perrin, C. (2010), Are seemingly physically similar catchments truly hydrologically similar? Water Resources Research, 46, W11558, doi:10.1029/2009WR008887

---

## Author Comment (AC4) · 26 May 2020

**Reviewer 3: Genevieve Ali**

COMMENTS

In this manuscript entitled "Future streamflow regime changes in the United States: assessment using functional classification", two main goals are pursued: (1) develop a catchment classification scheme for streamflow regimes, and (2) use this scheme to evaluate changes in future flow regimes. Contrary to the majority of previously published catchment classification efforts, here the authors decided not to rely on stream-flow indices. Instead, they are using a functional approach via which the shapes of mean annual hydrographs are classified, this in order to retain temporal autocorrelation information. Overall, the manuscript is of appropriate length, well written and with good-quality figures and tables. The dual focus of the manuscript on catchment classification and climate change impact assessment is very interesting, and I agree with the authors about their description of the advantages of functional classification. I did find that a few statements made in the manuscript warranted clarification, and that some details regarding the datasets, process interpretations, or linkages with existing literature were lacking (see specific comments below). With revisions, I believe that this manuscript will be interesting to the HESS readership, and a great addition to our growing body of literature on catchment classification.

**Reply:** *Thank you very much for your thorough review. We provided missing details on the dataset, added a few clarifications, and extended the discussion as suggested. The changes are discussed in detail below.*

SPECIFIC COMMENTS

N.B.: page and line numbers are noted as PX (page X) and LX (line X).

Section 2.1: Given the international readership of HESS, I think that more detailed in-formation is needed about the catchment selection criteria. For people not familiar with the CAMELS dataset, it is quite unclear what is meant by "minimum human impact": is the human impact assessed in terms of catchment-wide land use (that would mean no agricultural or urban catchment), or river regulation? And how may the answer to that question affect the generalization potential of the manuscript conclusions? In other words, the authors should discuss the limitations associated with not considering human-impacted catchments in the present study... Also, how was the 1981-2018 data period chosen for the analysis?

**Reply:** *Thank you for pointing out the need for clarification. We focused on catchments with minimum human impact to be able to look at the effects of climate change on flow regimes in isolation. The catchments belong to the HCDN-2009 network [Lins, 2012], which is a set of stations deemed suitable for analyzing hydrologic variations and trends in a climatic context. The dataset consists of catchments with natural flow conditions undisturbed by artificial diversions, storage, and other activities in the drainage basin or the stream channel and show less than 5% imperviousness as measured by the National Land Cover Database [Jin et al., 2013]. We added this information to the text. We also specified that the period 1981-2018 was chosen 'as data for this period was available for most stations in the dataset'.*
*As suggested, we added a discussion on the limitations of the classes in the case human-impacted catchments are of interest: 'The streamflow regime classes identified here do not*

*comprise classes of catchments with major flow alterations as the clustering was performed using streamflow regimes from catchments with minimal human impact. The five classes proposed here are therefore of limited use if a problem requires including catchments with strong human flow alterations. A flow regime of a regulated stream may still be attributed to one of the five regime classes identified if the altered regime shows similarities with the flow seasonality and variability of one of the 'natural' classes. However, if flow alteration leads to the emergence of regimes clearly distinct from those observed under natural conditions, additional regime classes would be necessary. In addition, the relationships between catchment characteristics and class memberships would need to be revised to enable the assignment of ungauged catchments to one of the classes in the updated set.'*

P5 L120: There is a reference to characteristics with missing values. Which characteristics (or types of characteristics) are the ones with missing values? Did omitting them lead to biased results?

**Reply:** *Among the 33 characteristics available, 2 had missing values (i.e. 'second most common geologic class in the catchment' and 'subsurface porosity'). They both belong to the class of geological characteristics comprising 7 characteristics in total, which means that we were still able to consider 5 characteristics related to geology. We found, however, that these geological characteristics were of minor importance for explaining regime class membership. We specified the two classes with missing values in the manuscript.*

Section 2.5, specifically L180-182: How was the comparison made, exactly, from a quantitative or statistical standpoint? Using contingency tables or crosstabs? Or something else? This is a bit unclear to me.... Maybe because I was expecting a statistical comparison when in fact, it is not what was done...

**Reply:** *We checked whether the predicted future class corresponded to the class of the reference simulation. The outcome of this check is binary: 0: predicted future class corresponds to reference class, 1: predicted future class differs from reference class. The results of this comparison are shown in Figure 6 (bars on the left). For the catchments with regime changes, we then identified the direction of change using a contingency table of counts (Figure 6, colored bars on the right). We specified in the text that: 'We then compare the predicted future classes to the class of the corresponding reference simulation using a contingency table of counts.'*

Figure 3: The different (graphical) features of the boxplots should probably be de-scribed in the figure caption. I assume that the horizontal black lines refer to the medians.... what do the whiskers represent, though: 1 interquartile range (IQR), 1.5 IQR, min and max values, or something else? Are there no statistical outliers associated with each cluster, i.e., each individual box?

**Reply:** *Thank you for pointing out the need for specification. We added the following text to the caption: 'The black lines in the boxplot indicate the median, the upper and lower whiskers correspond to $1.5 * R_{IQ}$, where $R_{IQ}$ is the inter-quartile range. Outliers are not displayed.'*

P9 L203-204: That should not be a surprise, given that the flood and drought definitions are hydrograph-based.... or am I missing something?

**Reply:** *The droughts and floods were determined using a threshold-level and a peak-over-threshold approach, respectively while the regime classes was determined using the mean annual*

*hydrographs where extremes are smoothed out. But yes, we would expect some correspondence between the streamflow regime of a catchment and the types of extreme events it experiences. We here show that flood and drought event characteristics of the different streamflow regime classes are indeed distinct (Figure 3 in the manuscript).*

P9-10, L207-210: The text description, here, does not underline that strong of a contrast between the weak winter regime and the strong winter regime. Maybe it can be rephrased for the contrast to be expressed more strongly?

**Reply:** *We added the following sentence highlighting the differences between catchments with a weak and strong winter regime: 'Compared to catchments with a weak winter regime, catchments with a strong winter regime lie at higher elevations, show higher fractions of snow and are characterized by larger flood magnitudes.'*

P10 L217-218: That would explain why there is such a large degree of spatial contiguity/spatial autocorrelation within each cluster. However, it is a bit unclear to me, from the text, whether a RF classification using climatological variables only performs equally as well as – or better than – a RF classification that used both climatological and physiographic variables.

**Reply:** *You are right, we did not discuss whether the random forest model profits from including additional physiographical characteristics in addition to climatological ones. We just discussed variable importance in the context of the 'full model' including all potential explanatory variables. If physiographical variables are excluded from the random forest model, the prediction error increases from 10% to 12%, which corresponds to a marginal decrease in model performance. So yes, a random forest classification using climatological variables performs almost equally as well as a model also including physiographical variables. We added the following sentence to the text: 'Excluding these physiographical explanatory variables from the random forest model results in only a small decrease in prediction performance (prediction error 12%).'*

P10 227: The authors stated that "However, our clustering scheme avoids the formation of very small clusters seen in Jehn et al. (2019)." First, what might explain this? Second, the authors seem to imply that having very small clusters is an inconvenient, and I am not sure I agree – very small clusters could represent very local conditions or hotspots, which are real. The authors should either rephrase or at least nuance their statement to clarify what they mean.

**Reply:** *We did not intend to suggest that forming small clusters is necessarily a bad thing and therefore reformulated the sentence using neutral wording:' However, our clustering scheme results in larger clusters than the ones seen in Jehn et al. (2019).' This is mostly related to the fact that we chose to work with fewer clusters. If we further increased the number of clusters to e.g. 7 instead of 5 clusters (Figure 1 in this response to the reviewer), we would also introduce very small clusters. We would further split up the melt-regime cluster and the New-Year's-regime clusters. This does, however, not further improve cluster distinctiveness as measured by the mean silhouette width.*

[Figure]

*Figure 1: Map of 671 catchments in the dataset clustered into 7 streamflow regime classes. Each color represents a different class.*

P10 L230-234: The authors wrote that "The strong link between regime classes and meteorological and physiographical catchment characteristics allows for the attribution of ungauged catchments, where streamflow data are not available, to one of the regime classes, which is potentially very useful for the prediction of streamflow characteristics in ungauged basins". I am not sure where that statement is coming from, as ungauged catchments were not examined in the present study. I agree that the present study might have interesting implications for predictions in ungauged catchments, but this statement, as written, reads as a result when in fact it is an interpretation. In the same line of thought, I wonder whether it would be possible to have separate Results and Discussion sections in the manuscript. There are a few instances, in the text, where it can be tricky to distinguish whether a plain result/fact is being stated, or whether a hypothesis/interpretation is being put forward.

**Reply:** *We split up the Results and Discussion section into two sections in to more clearly distinguish between the results of our study and their implications. It is correct that the focus of our study is not on prediction in ungauged basins. However, we show that a random forest model fitted to climatological and physiographical characteristics is well able to attribute a catchment to one of the regime classes without having any information on streamflow (class prediction error 10%, see l. 118-124 and l.215-119). We add the following sentence to the methods section: 'To further investigate the physiographical and climatological controls on regime class membership and to check whether regime classes can potentially be predicted for ungauged catchments, we perform a random forest classification'. Thanks to its low prediction error, this random forest model enables attributing of ungauged catchments to one of the regime classes. As we do not go into detail on this aspect, we moved the statement to the new Discussion section and clarified it as follows: 'The strong link between regime classes and meteorological and physiographical catchment characteristics enables attributing ungauged catchments, where streamflow data are not available, to one of the regime classes. This attribution can be achieved by using the first random forest model fitted in this analysis enabling predictions of regime class membership using physiographical and climatological characteristics. The ability to attribute of an ungauged catchment to one of the regime classes is potentially very useful to predict of streamflow characteristics in ungauged basins.'*

Figure 4: This figure is quite interesting but the comparison of "climate sensitivity" between observations and simulations appears quite qualitative. I wonder: 1) How were the five example

catchments showcased in this figure chosen (or, are those sites representative of median cluster conditions)?; and 2) Was a quantitative method of comparison between observations and simulations used for all catchments?

**Reply:** *We chose one regime per cluster and the sites do not necessarily represent median cluster conditions. Yes, we also applied a quantitative method to evaluate 'climate sensitivity' over all catchments (l.241-243). We added that: 'The sensitivity gradients are computed on the response surface of each catchment in the horizontal direction for temperature and in the vertical direction for precipitation.' The results of this quantitative evaluation are summarized in Figure 2 in this response to the reviewers. The statement: 'Higher mean precipitation leads to higher mean discharge independent of the catchment and regime. The reaction of streamflow to temperature, however, seems to depend on the catchment because the relationship between mean temperature and mean discharge is generally weak and can be positive or negative. (l.238-240)' can therefore be generalized to the entire dataset. We preferred to show the sensitivity grids for a few catchments as we think that these examples nicely illustrate the mechanisms we see for the whole dataset.*

[Figure]

*Figure 2: Observed vs. simulated sensitivity gradients for temperature (left) and precipitation (right) over all catchments computed using climate sensitivity grids as displayed in Figure 4 of the manuscript for five example catchments.*

P11 L240: The authors refer to a "visual analysis"; were all plots for all 605 catchments visually analyzed?

**Reply:** *We computed such sensitivity grids for each catchment and used them to compute horizontal and vertical sensitivity gradient as outlined in the response to the previous question.*

P11 L243-244: The Methods section should explicitly state what the Kolmogorov-Smirnov test was used for, the assumptions being it, and the null and alternate hypotheses (so that readers know what the test results mean). Also, a test cannot be rejected: we can only reject or fail to reject a null hypothesis, so that sentence should be reworded.

**Reply:** *We rephrased the sentence to: '(Kolmogorov-Smirnov test does not reject the null hypothesis that observed and simulated gradients were drawn from the same continuous distribution at level of significance alpha=0.05.)'*

Figure 5: Lines are a bit difficult to distinguish on this figure; making it larger and changing the symbology might help.

**Reply:** *Thank you for your suggestion. We rearranged the plot into a 3 rows, 2 columns format to increase the size of the individual subplots. In addition, we darkened the color of the control regime to increase contrast with respect to the regimes simulated using the GCM output.*

P12 L258-259: The authors wrote "In contrast, regimes with a strong seasonality such as strong winter and New Year's regimes are well simulated". What about the melt regime, which is also highly seasonal?

**Reply:** *This statement is also valid for melt regimes and we added this regime type to the list.*

Figure 7: If the black circles mean no regime change, the legend should state so.
**Reply:** *Yes, black circles refer to no regime changes. We added this to the legend of Figure 7.*

COMMENTS SPECIFIC TO DISCUSSION ELEMENTS WORTH INCLUDING IN THE MANUSCRIPT

Discussion comment #1: In the present study, regime clusters appear equivalent to clusters derived based on physiographic similarity and clusters derived based on climatological similarity... this is contrary to studies published by Ali et al. (2012) and Oudin et al. (2010) – in a comforting way, I might add – and this should probably be discussed. The "overlap" or agreement between the different classifications bodes well for using climatic and physiographic information as a proxy for streamflow regime types. The fact that an agreement was found in the present study and not in others may be due to the fact that here, functional data were used instead of select streamflow indices.

**Reply:** *Thank you for suggesting to expand the discussion on this aspect. We added the following discussion point: 'We find functional data clustering to be a useful tool for identifying clusters of catchments with not only similar streamflow regimes but also similar catchment, meteorological, flood and drought characteristics. This similarity corroborates findings by Bower et al.* (2004) *and McCabe and Wolock* (2014) *who established a clear link between similarity in streamflow seasonality and climatic and physical similarity. However, it is in contrast to findings by Ali et al.* (2012) *who found that catchments similar with respect to a set of flow indices are not necessarily physically similar. Explicitly including seasonality or information on the temporal autocorrelation of regimes may therefore help to identify clusters of catchments which are not only hydrologically but also physically similar.' A reference to Oudin et al.* (2010) *was added to the introduction.*

Discussion comment #2: It is not a study limitation per se, but the authors may want to discuss the rationale for using functional streamflow data classification (to preserve temporal information) while NOT using climate time series (e.g., mean annual hyetograph) for classification purposes. When I started reading the manuscript, I was puzzled by the fact that a classification based on temporally autocorrelated data (i.e., whole annual hydrographs) was going to be compared to a classification based on climate indices. In other words, I wondered how the analyses would turn out given that different regions may have similar values of mean annual precipitation, even though the temporal distribution of that precipitation may be skewed in some places but not elsewhere. In the end, the authors found that they could neglect the temporal information included in climate time series and still manage to use that climate information (i.e., the climate index class) as a good proxy for streamflow regime class (which, itself, is based on temporally autocorrelated data). That warrants discussion, I think, as it is a bit counter-intuitive (to me, anyway...)

**Reply:** *Our functional streamflow regime clustering approach is indeed solely based on the mean annual hydrographs and the temporal autocorrelation contained therein. It does not rely on climate time series. The information on climate characteristics is only used to see whether the hydrological regime clusters are also climatologically meaningful. We clarify this in the introduction by saying: 'This scheme makes better use of the seasonal and temporal information stored in the hydrological regime than index-based approaches and is solely based on streamflow information (i.e. no climatological information is used).' We indeed find that these clusters formed according to mean annual hydrographs are distinct in terms of climate and physiographical characteristics (Figure 3 in the manuscript). The good predictive power of a random forest model in correctly attributing catchments to a regime cluster based on climate and physiographical characteristics supports this (l.215-217).*

Discussion comment #3: The authors may want to use the concepts of resistance, resilience and synchronicity discussed by Carey et al. (2010): those concepts partly echo what the authors are referring to as "climate sensitivity".

**Reply:** *Thank you for this suggestion. We extend the introduction to the climate sensitivity analysis as follows: 'In the climate sensitivity analysis, we assess whether the hydrological model reacts to changes in mean temperature and precipitation in the same way as observations. In terms of precipitation, this corresponds to checking whether the model captures the resistance of a catchment, i.e. the degree to which runoff is coupled with precipitation Carey et al. (2010).'*

EDITORIAL SUGGESTIONS
P2 L30: "illustrate the hydrological functioning" seems more appropriate than "govern the hydrological functioning", since the authors are referring to streamflow regimes.
P2 L31: I think that the phrase "influencing streamflow variability" should be changed....
Otherwise the whole sentence read as "The characteristics of streamflow regimes [influence] streamflow variability and seasonality", which reads as a circular statement.

**Reply:** *We rephrased this sentence to: 'The characteristics of streamflow regimes, as described here by mean annual hydrographs, include streamflow variability and seasonality and influence the hydrological functioning of a catchment.'*

P10 L217: "shows that the the most important variables for" SHOULD BE CHANGED FOR "shows that the most important variables for"
**Reply:** *We eliminated the duplicate 'the'.*

P11 L243: "Klomogorov–Smirnov" SHOULD BE CHANGED FOR "Kolmogorov-Smirnov"
**Reply:** *We fixed this typo.*

P13 L274: "In contract" SHOULD BE CHANGED FOR "In contrast"
**Reply:** *We fixed this typo.*

REFERENCES CITED IN THIS REVIEW
- Ali, G., Tetzlaff, D., Soulsby, C., McDonnell, J. J., and Capell, R. (2012), A comparison of similarity indices for catchment classification using a cross-regional dataset. Advances in Water Resources, 40, 11-22. doi:10.1016/j.advwatres.2012.01.008
- Carey, S.K., Tetzlaff, D., Seibert, J., Soulsby, C., Buttle, J., Laudon, H., McDon-nell, J., McGuire, K., Caissie, D., Shanley, J., Kennedy, M., Devito, K. and Pomeroy,J.W. (2010),

Intercomparison of hydroclimatic regimes across northern catchments: synchronicity, resistance and resilience. Hydrological Processes, 24: 3591-3602. doi:10.1002/hyp.7880

- Oudin, L., Kay, A., Andréassian, V., and Perrin, C. (2010), Are seemingly physically similar catchments truly hydrologically similar? Water Resources Research, 46, W11558,doi:10.1029/2009WR008887

**References used in this response to the reviewer**

Ali, G., D. Tetzlaff, C. Soulsby, J. J. McDonnell, and R. Capell (2012), A comparison of similarity indices for catchment classification using a cross-regional dataset, *Adv. Water Resour.*, *40*, 11–22, doi:10.1016/j.advwatres.2012.01.008.

Bower, D., D. M. Hannah, and G. R. McGregor (2004), Techniques for assessing the climatic sensitivity of river flow regimes, *Hydrol. Process.*, *18*(13), 2515–2543, doi:10.1002/hyp.1479.

Carey, S. K. et al. (2010), Inter-comparison of hydro-climatic regimes across northern catchments: Synchronicity, resistance and resilience, *Hydrol. Process.*, *24*(24), 3591–3602, doi:10.1002/hyp.7880.

Jin, S., L. Yang, P. Danielson, C. Homer, J. Fry, and G. Xian (2013), A comprehensive change detection method for updating the National Land Cover Database to circa 2011, *Remote Sens. Environ.*, *132*, 159–175, doi:10.1016/j.rse.2013.01.012.

Lins, H. F. (2012), *USGS Hydro-Climatic Data Network 2009 (HCDN–2009): U.S. Geological Survey Fact Sheet 2012–3047*, Reston, VA.

McCabe, G. J., and D. M. Wolock (2014), Spatial and temporal patterns in conterminous United States streamflow characteristics, *Geophys. Res. Lett.*, *41*(19), 6889–6897, doi:10.1002/2014GL061980.